# Subnational variations in the quality of household survey data in sub-Saharan Africa

Valentin Seidler [1], Edson C. Utazi [2], Amelia B. Finaret [3,4], Sebastian Luckeneder [5], Gregor Zens [6], Maksym Bodarenko [2], Abigail W. Smith[4], Sarah E. K. Bradley[7], Andrew J. Tatem [2] & Patrick Webb [8]

Nationally representative household surveys collect geocoded data that are vital to tackling health and other development challenges in sub-Saharan Africa. Scholars and practitioners generally assume uniform data quality but subnational variation of errors in household data has never been investigated at high spatial resolution. Here, we explore within-country variation in the quality of most recent household surveys for 35 African countries at 5 × 5 km resolution and district levels. Findings show a striking heterogeneity in the subnational distribution of sampling and measurement errors. Data quality degrades with greater distance from settlements, and missing data as well as imprecision of estimates add to quality problems that can result in vulnerable remote populations receiving less than optimal services and needed resources. Our easy-to-access geospatial estimates of survey data quality highlight the need to invest in better targeting of household surveys in remote areas.

The absence of dependable data on age, fertility, mortality, health, wealth, education, and nutrition in many low-income countries has been a long-standing concern for scientists and practitioners of human development[1,2]. Lacking reliable and representative census data, public health information, and/or birth registration systems, most African countries rely on multi-purpose household surveys for information on their citizen's well-being[3–5]. The most widely used sources of nationally-representative data by practitioners and the research community are the Demographic and Health Surveys (DHS), Multiple Indicator Cluster Surveys (MICS), and Living Standards Measurement Surveys (LSMS). DHS data, the main example used in this study, have been used in about 6000 publications since 2010 (a conservative estimate)[6].

Today, independent researchers and the DHS program routinely use geocoded household survey data to produce high-resolution modelled surfaces of health and development indicators[7–14] often in conjunction with spatial data from satellite images and other secondary data sources, such as data on armed conflict[15–18] or disease

distribution data[19]. Other household survey programs are now following their lead, e.g., MICS[20]. Since policy decisions can take place at national or subnational levels, and the assessment of impacts is usually pursued at district level or lower, there is an increasing demand for small area estimates[21]. For example, the World Health Organization (WHO) frequently uses estimates from the Malaria Atlas Project (MAP)[19,22], which constructs modelled surfaces of malaria prevalence from geocoded household survey data[23]. Moreover, GAVI, the Vaccine Alliance, and the Nigerian government are making use of such high-resolution modelled estimates to guide strategies around reaching zero-dose children[24,25], while the Children's Investment Fund Foundation (CIFF) are utilizing these approaches for monitoring, evaluation and learning applications[26].

The spatial dimension of policymaking critically depends on the quality of underlying sub-national and regional data. However, the quality of household survey data across sub-national regions cannot be assumed to be uniform. Nutrition and public health professionals as well as government ministers have called for fixing data gaps and

[1]Central European University, Vienna, Austria. [2]WorldPop, School of Geography and Environmental Sciencevundefined, University of Southampton, Southampton, UK. [3]University of Edinburgh, Global Academy of Agriculture and Food Systems, Edinburgh, Scotland, UK. [4]Allegheny College, Department of Global Health Studies, Meadville, PA, USA. [5]Vienna University of Economics and Business, Department of Socioeconomics, Vienna, Austria. [6]International Institute for Applied Systems Analysis (IIASA), Laxenburg, Austria. [7]Independent Public Health Demographer, Washington D.C., USA. [8]Tufts University, Friedman School of Nutrition Science and Policy, Boston, Massachusetts, USA. ✉e-mail: seidlerv@ceu.edu

improving data quality[7,27]. Illness and malnutrition are defined as quantified conditions lying outside 'healthy' thresholds. Thus even modest inaccuracies, such as errors and artifacts in the measurement of age can give a false picture of, for example, child stunting (low height for age)[28–31]. Robust subnational information, on the other hand, enables health authorities to target and better fund health interventions[32].

We define high quality data as having low statistical uncertainty and little systematic measurement bias and systematically missing data. Statistical uncertainty is mainly determined by sampling strategy (determining for example sample sizes) and can occur independently of systematic measurement error and missing data. While high statistical uncertainty typically implies decreasing predictive power by widening prediction intervals, taking into account statistical uncertainty is readily achieved in standard statistical frameworks for empirical analysis, for example using stochastic error terms. Systematic measurement error and missingness are more serious threats to the derivation of empirical conclusions. These data errors cannot be accounted for because their magnitude and spatial distribution are very often not known[12,26,33–35]. The DHS program and independent researchers have examined measurement errors and missing data in household survey data on aggregate levels only - across countries and across survey teams within countries[28,36–41]. This is unsatisfying as no quantification of spatial variation in household survey data quality exists for a large number of countries at highly granular levels such as the village level. As a result, even those practitioners and researchers who work extensively with national household surveys and who are aware of the potential for spatial variation in data quality, may not know the precise spatial distribution and the magnitude of measurement errors and missing data.

These data errors can occur despite rigorous efforts and enumerator training[42–44]. Errors may be introduced at any stage of the household survey beginning with survey design[45], during data collection or data processing[38]. Existing studies point to a number of variables such as the age of respondents[46], low numeracy and literacy or lack of birth registration[30] that may be less reliable in remote areas relative to urban areas, but we lack spatial estimates of data errors for specific locations, which in turn adds to the already greater vulnerability of the so-called last-mile populations in low-income countries (defined by The World Bank as having less than $1145 gross national income per capita in 2023)[37,47]. Beyond the fields of health and demography, which are the focus of this study, development and agricultural economists have been working towards high-quality economic and health data from low- and middle-income countries (defined by The World Bank as having less than $4515 in gross national income per capita in 2023), and have improved data quality through several strategies, including by incorporating spatial and community-level data, using GPS technology, or engaging with local communities (Supplementary Note 1)[48–51]. Similarly, the DHS program and international donors made significant efforts to ensure high data quality by identifying potential sources of data errors[36,42,52–55] and by making continuous improvements[41,56–59].

The precise spatial distribution of measurement errors and missing data has not been quantified before at subnational level for a large group of countries[41,43,59–62]. Not knowing the spatial variation in these data errors is unsatisfactory, because decisions are increasingly made locally, for example at the level of districts, and because so many decisions are made on the assumption that local data are reliable. In addition, researchers and other data users who do not frequently working with household data may not be aware of, or simply ignore, the potential for spatial variation in the quality of data with which they are working[63]. Even researchers and practitioners who work extensively with DHS data, and who are aware of the potential for data errors, can improve causal inference with knowledge of the extent of variation in data quality in a particular region. Finally, the DHS program

has highlighted the need for greater attention to spatial variation in data quality. Their goal is to improve the performance of survey teams working in locations prone to high data errors by providing them with, for example, better training or more resources[37].

In this paper, we provide a high-resolution geospatial analysis of DHS data quality taken from surveys of central, eastern, western, and southern Africa between 2006 and 2022. Among large household surveys in Africa, the DHS program offered the most complete data covering 35 of Africa's 46 countries. The number of DHS survey rounds varies across nations. In countries with more than one survey round since 2006, the most recent survey was used (Supplementary Table 9).

From this set of household surveys, we derived three data quality indicators which are widely used by practitioners and researchers (Supplementary Note 2). 'Incomplete age' refers to the share of interviewed women (15–49 years) with either the year or the month of birth missing relative to all interviewed women. 'Age heaping' refers to the proportion of reported ages ending in 5 or 0 of all adults between 23 and 62. This is based on Whipple's index definition, a value above 20% indicates age heaping. Finally, 'flagged HAZ' refers to missing or biologically implausible values for the measured heights of children (height-for-age z-scores) according to WHO (World Health Organization) standards.

In this work, we map predictions of data errors beyond data points in the analyzed DHS datasets. To do this, we employ Bayesian model-based geostatistics that combine spatially explicit data and covariates from gridded high-resolution datasets to produce $5 \times 5$ km gridded estimates of the three data quality indicators - a level of detail that is roughly equivalent to the size of villages in rural areas. We aggregate our estimates to districts and to national levels using population weights. We provide an online data visualization tool to provide easy access to our estimates for data users (https://apps.worldpop.org/SSA/data_quality/). Next, we combine original DHS data with freely available data on settlements and nighttime light emissions to explore non-random distribution patterns of data errors and chart the deterioration of DHS data quality with increasing distance to settlements. Finally, we contrast the spatial distribution of our predictions of data errors with the statistical uncertainties associated with related public health indicators in the DHS data, which may collectively threaten the quality of insights and decisions that can be drawn from DHS data.

## Results
### Subnational variability in data quality
Predictions of data quality varied substantially across $5 \times 5$ km cell levels and across district levels throughout the 35 included African countries. Within-country variation was apparent with all three data quality indicators for all 35 included countries (Fig. 1) and it was higher than what would be the product of chance. Our data quality predictions were obtained from a geostatistical model trained on age heaping, incomplete age and flagged HAZ derived from DHS data (see Methods for details). We provide a description of these three data quality indicators in the Supplementary Note 2. $5 \times 5$ km cell levels of geographical resolution are commonly used in studies using DHS data and by the DHS program[8,10,11,13,14,64].

The Moran's I statistic, a measure of spatial autocorrelation that we computed using raw DHS data at the district level, was recognizable and varied in value across countries and across our three indicators. Spatial autocorrelation at the district level was more identifiable for incomplete information on women's age while the subnational variation in age heaping and flagged HAZ values appeared closer to random (Supplementary Tables 10–12).

Within-country variation in predictions of data quality was of larger magnitude in countries with lower average data quality (Supplementary Fig. 2). For example, our estimates of age heaping in Nigeria (national mean 39.8%) ranged from 62.1% (standard deviation-

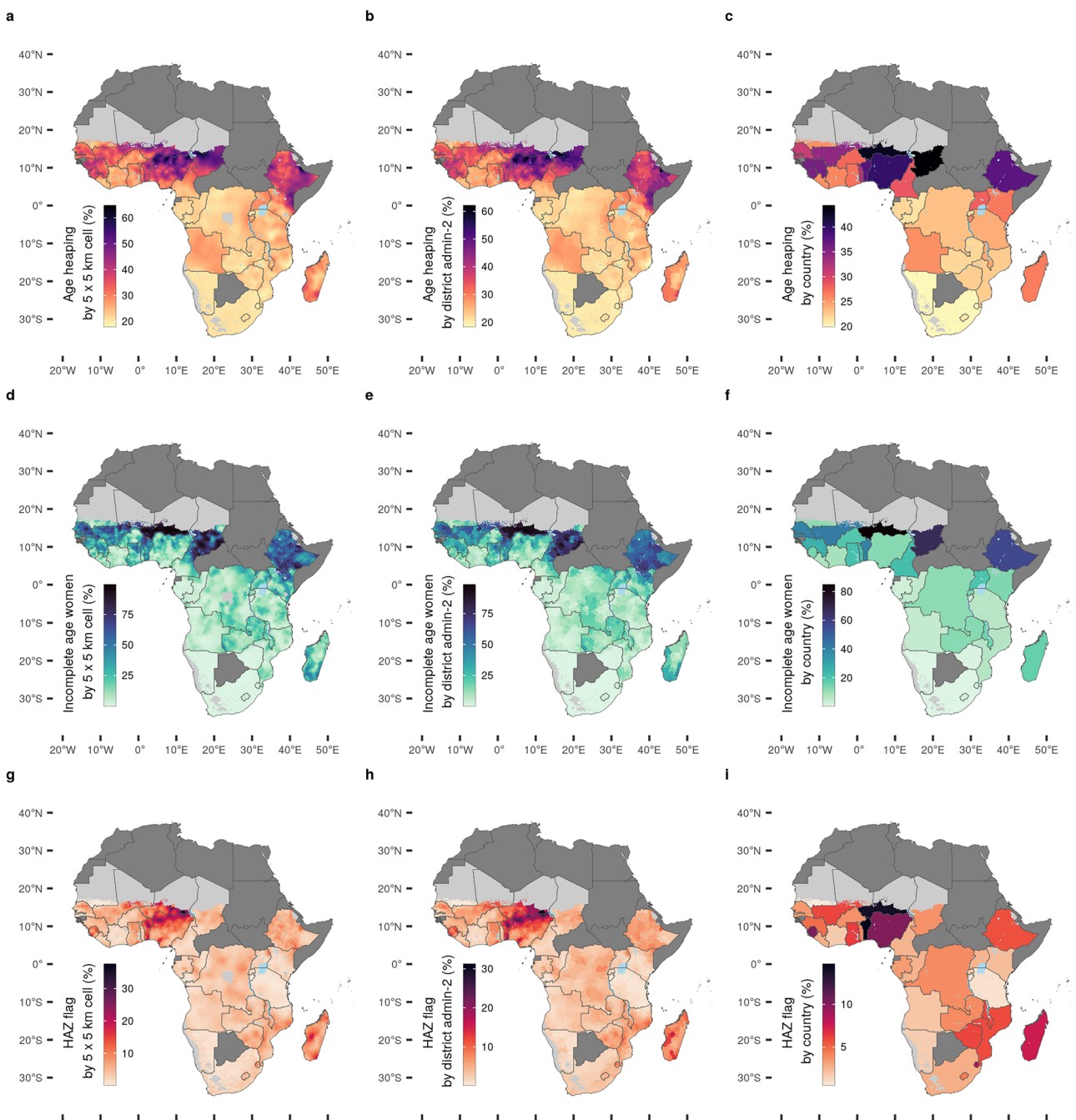

**Fig. 1 | Distribution of measurement errors in Demographic and Health Surveys (DHS) data 2009-2022. a–i** Proportion of reported ages ending in 5 or 0 of all adults between 23 and 62 ('age heaping') at (**a**) 5 × 5 km grid-cell level; (**b**) district level (admin-2); (**c**) country level. Share of interviewed women (15–49 years) with either the year or month of birth missing relative to all interviewed women ('incomplete age') at (**d**) 5 × 5 km grid-cell level; (**e**) district level (admin-2); (**f**) country level. Missing or biologically implausible values for the attained heights of children (height-for-age z-scores, HAZ) according to World Health Organization (WHO) standards ('flagged HAZ') at (**g**) 5 × 5 km grid-cell level; (**h**) district level (admin-2); (**i**) country level. Countries in dark grey are not in the sample. Grid cells with fewer than 10 people per 1 × 1 km and classified as barren or sparsely vegetated or grid cells with population data not available are colored light grey[83,94–96].

sd, 2.2%) in Danmusa district in Katsina state in northwest Nigeria to 25.4% (sd 5.6%) in Agege in Lagos State, where the terminal digit preference for 0 or 5 was just 5.4 percentage points above the expected natural occurrence of 20%. Estimates of the share of incomplete information on women's age in Chad (national mean 67.3%) ranged from 91.6% (sd 2.7%) in Loug Chari in the south of Chad to only 8.1% (sd 2.4%) in Dar Tama, on the eastern border with Sudan. Flagged HAZ values in Madagascar (national mean 7.58%) ranged from 14.8% (sd 1.9%) in Bongolava to 4.5% (sd 0.95%) in Haute Matsiatra, two regions in the center of the island. These estimates demonstrate that analyzing

systematic measurement errors at the country-level alone masks important local and regional patterns

There was little to moderately positive correlation between the three data quality indicators. The correlation coefficients between pairs of indicators ranged from 0.35 between age heaping and incomplete age to 0.71 between the two age-related indicators, age heaping and flagged HAZ (Supplementary Fig. 3). Data quality challenges are not uniform across the indicators we employed, and different correlation coefficients reflected different underlying potentials for systematic measurement error and missing data.

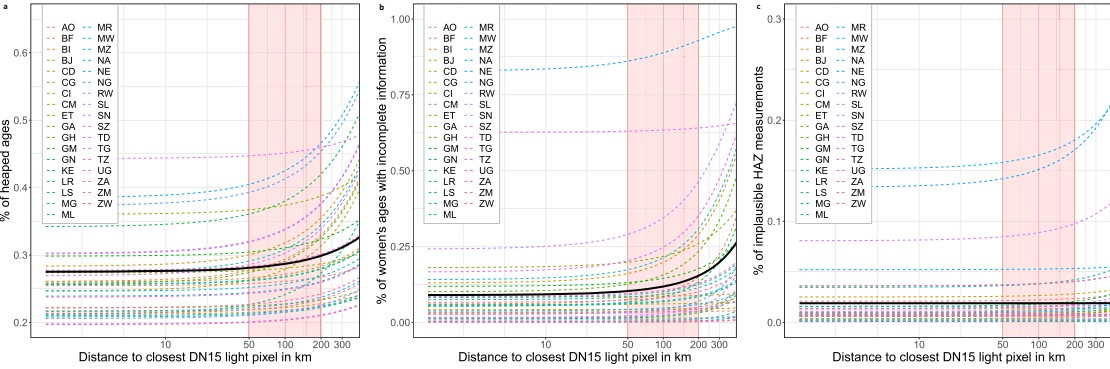

**Fig. 2 | Predicted data quality by distance to closest Digital Number (DN) 15 nighttime light emitting source in km (logarithmic scale). a–c** Predictions obtained from regional binomial logistic regressions on distance (in km) to closest DN 15 light pixel of (**a**) share of reported ages ending in 5 or 0 of all adults between 23 and 62 ('age heaping'), (**b**) share of interviewed women (15–49 years) with either the year or month of birth reported missing ('incomplete age'), and (**c**) implausible or missing values for the attained height-for-age z-scores (HAZ) of children under five according to World Health Organization (WHO) standards ('flagged HAZ'). All models include country fixed effects.

The only moderate and low correlation between the three data quality indicators reflected a distinctive subnational spatial pattern for each indicator. This suggests varying degrees of difficulty encountered in collecting the different types of measurements across localities and across DHS data collection teams[37]. For example, incomplete age records reported from Wajir West (53.3%, sd 5.9%) in northern Kenya were considerably worse than the national mean of 13.4% while in the same district flagged HAZ values (4.0%, sd 0.94%) were closer to the Kenyan national mean of 2.2%.

### Deteriorating data quality in remote areas

To explore the potential of a collection bias in the spatial distribution of measurement and missing data, we combined the prevalence of errors by survey location in the raw DHS household data with distance to the nearest settlements. We conducted the analysis for each of the three data quality indicators to determine the extent to which they were subject to a remoteness penalty, that is, an increase in the prevalence of measurement errors, artifacts, and missingness with distance to settlements. Remoteness captures a mix of potential factors such as respondent literacy and numeracy, or more challenging conditions for survey collection teams. We did not intend to study these underlying causes. Our aim was to explore the magnitude of this problem and whether it affected the three data quality indicators to a similar extent. In terms of further empirical analysis, the data challenges systematically increasing with remoteness could threaten the insights drawn from the data in two ways. In the best case, measurement errors and missingness are random, leading to more noisy estimates; in the worst case, errors and missingness are systematic, leading to systematic bias in parameter estimates. Data users will need to assess on a case-by-case basis which of these two threats is more relevant to their study based on geographic location, point in time, and variables of interest.

The predictions underlying Fig. 2 were based on nighttime light (NTL) emissions of a specific luminosity (Digital Number, DN 15), which lies above the threshold of the luminosity of public streetlights indicating a settlement with sufficient infrastructure for survey teams to restock equipment and rest[65–67]. However, the differences in data quality were similarly apparent with lower and brighter light emissions threshold levels (Supplementary Figs. 5, 6). The deterioration in data quality was also discernible with distance to settlements using two alternative high-resolution settlement data sets built from daylight and radar satellite imagery and electoral records. These alternative data sets include smaller and unlit settlements with shorter geographical distances between them, which made this approach less informative relative to using NTL emissions but displayed similar trends across all

three data quality indicators (Supplementary Figs. 7, 8)[68]. Finally, we found little evidence that the observed bias was based on one-time effects for the majority of the included countries with multiple survey rounds (Supplementary Fig. 9–11). Data from countries with more than one survey round since 2006 indicate that the spatial bias is a long-term problem for the large majority of the 24 countries with multiple survey rounds.

In the 35 included African countries, increasing distance from electrified settlements emitting NTL of DN 15 was associated with decreasing data quality across all regions. The differences in quality were relatively worse across West Africa, and relatively better in Central and Southern Africa (Supplementary Fig. 4). The bias affected all three data quality indicators. The bias was relatively stronger with age heaping (Fig. 2a) and incomplete age records of women (Fig. 2b). It is relatively less apparent for flagged HAZ values (Fig. 2c) which was consistent with cross-sectional studies specifically examining stunting[47]. For age heaping and incomplete age of women in particular, this association grew stronger within countries of medium to high overall levels of measurement errors (Supplementary Figs. 12–14). For example, the estimated share of mothers (15–49 years) with either the year or month of their infant's birth missing ('incomplete age') in Togo (national mean 22.3%) was 20.4% at 50 km distance, increasing to 24.8% at 100 km and was 35.3% at 200 km. The estimated proportion of adult ages ending in 5 or 0 ('age heaping') increased within Kenya (national mean 26.7%) from 26.2% at 50 km from the nearest town or nighttime light emitting area to 26.7% at 100 km and 27.9% at 200 km distance. The differences in data quality were smallest within countries that had overall higher quality DHS data. For example, age heaping within South Africa (national mean 20.2%) was at 20.0% at 50 km distance to the nearest point of nighttime light of DN 15 value. It was 20.3% at 100 km distance and 21.0% at 200 km.

### Data errors and sampling uncertainty

Figure 3 shows the spatial distribution of two of our estimates for data quality: 'incomplete age' and 'flagged HAZ' (in shades of blue), and it pairs them with the standard deviations of predicted estimates of related public health indicators: 'contraceptive use' and 'stunted children' (in shades of green). These indicators are related because contraceptive use is an important health indicator for women, as it relates to their overall health and ability to achieve their desired fertility. In addition, maternal health status is a major determinant of stunting. Standard deviations indicate statistical uncertainty, for example regarding the precision of estimates for the prevalence of contraceptive use among sexually active women ('contraceptive use') or stunting prevalence among children ('stunted children'). High standard deviations are mainly the result of small sample sizes at the survey

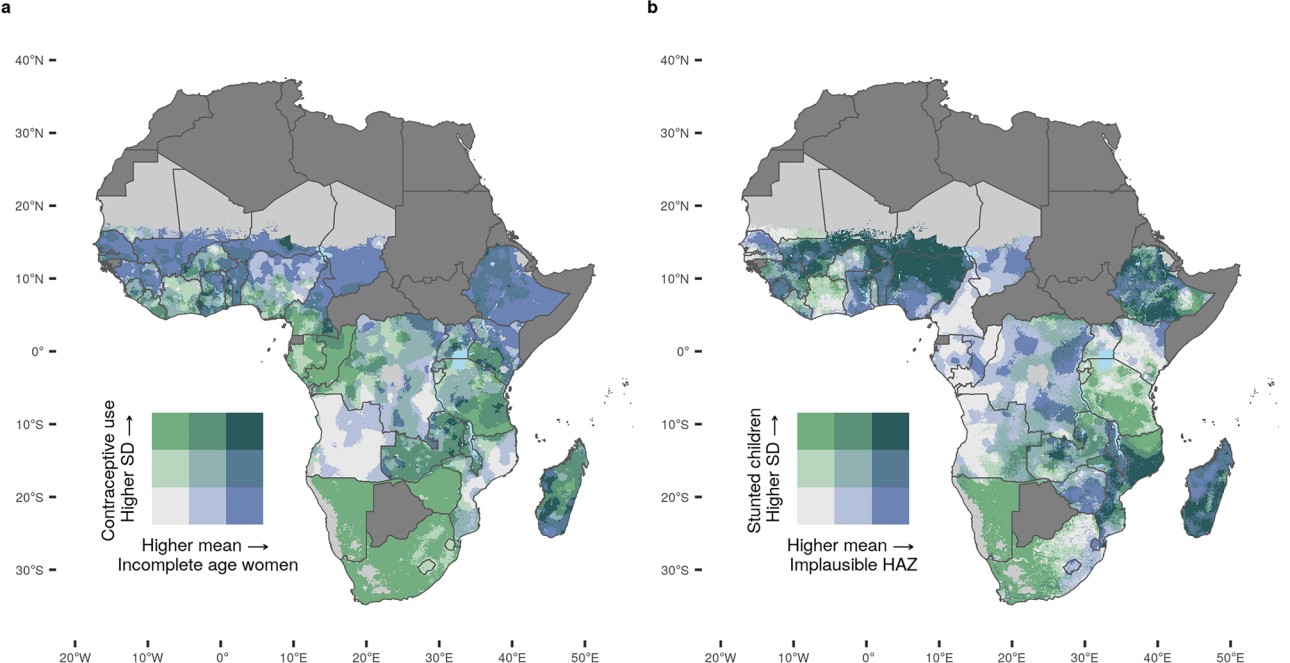

**Fig. 3 | Distribution of measurement errors and uncertainty of predicted estimates of public health indicators in Demographic and Health Surveys (DHS) data 2009–2022. a, b** standard deviations of predicted estimates of contraceptive use of sexually active women (in green) and incomplete age values (in blue) at 5 × 5 km grid-cell level (**a**), standard deviations of predicted estimates of stunting prevalence among children (in green) and flagged height-for-age (HAZ) values (in blue) at 5 × 5 km grid-cell level (**b**). Countries in dark grey were not in the sample. Grid cells with fewer than 10 people per 1 × 1 km and classified as barren or sparsely vegetated or grid cells with population data not available are colored light grey[83,94–96].

cluster level or from a distribution of survey locations that does not take into account geostatistical design considerations[12]. High uncertainties may even be consistent with the DHS sampling strategy, for example, to make economical use of resources. In terms of data quality, high uncertainties arising from small samples can for example reduce the predictive power of fitted models, but they are generally considered a lesser threat compared to systematic measurement bias and systematically missing data because they are typically directly accounted for in statistical modelling frameworks[12,69–71]. With only information on statistical uncertainty - i.e., without spatial quantifications of data errors such as those provided by our study - users of DHS data may be inclined to limit their assessment of data quality to easy-to-measure statistical imprecision and neglect the threat of measurement errors because they lack subnational information on the extent to which these two types of data problems overlap in the geographic area of their interest. By pairing for example, the sampling uncertainty of 'stunted children' with our estimates of missing or biologically implausible data ('flagged HAZ'), we inform DHS data users in which localities these two types of data problems overlap. Our findings can also help inform the DHS program where resources could be allocated to maximize data quality. This includes for example training enumerators, improving tools and procedures, or considering a larger sample size or new sampling strategy.

We estimated correlation coefficients at the district level (Supplementary Fig. 15). Missing data for women's age was moderately negatively correlated with the standard deviation of the estimated share of contraception users ($\rho = -0.402$, $p < 0.0001$). Flagged HAZ values were slightly negatively correlated with the standard deviation of the estimated prevalence of stunting ($\rho = -0.109$, $p < 0.0001$).

Some regions were challenged with either high standard deviations of both 'contraceptive use' and 'stunting' (for example in Namibia, Nigeria, parts of the Democratic Republic of Congo, Malawi, parts of South Africa and parts of Tanzania) or with high estimates of both 'incomplete age' and 'flagged HAZ' values (for example in Angola, parts of Democratic Republic of the Congo, parts of Ethiopia, parts of Kenya,

and Chad). On the country level, measurement errors paired with large sampling uncertainty were widely present only in Madagascar (national mean 'incomplete age' 16.3%, national mean 'flagged HAZ' 7.6%) and in Niger (national mean 'incomplete age' 84.6%, national mean 'flagged HAZ' 13.9%). On the local level, single districts were challenged by both high error estimates and large standard deviations of estimates. N'gauma district in north-western Mozambique reported estimates for 'incomplete age' of 21.5% and a standard deviation of 'contraceptive use' at 2.7% (mean 14.6%). In the same district, the estimate of 'flagged HAZ' was 6.6% and the standard deviation of 'stunted children' was 3.5% (mean 45.6%). The Mozambiquan national means of 'incomplete age' and 'flagged HAZ' were 6.7% and 5.8%, respectively. No single country exhibited low rates of errors and high statistical certainty nationwide. However, districts with high data quality appeared in clusters across the subcontinent. One example was several districts in Uíge Province in Northwest Angola (national mean of 'incomplete age' 4.4%), including Songo ('incomplete age' 0.5%, standard deviation of 'contraceptive use' 1.63), Buengas ('incomplete age' 0.6%, standard deviation of 'contraceptive use' 1.3), and Damba ('incomplete age' 0.6%, standard deviation of 'contraceptive use' 1.4). Similarly, near error-free HAZ values and high statistical certainty concentrated in districts surrounding Thiès such as Mbour ('flagged HAZ' 2.8%, standard deviation of 'stunted children' 1.02% (mean 15.3%) in western Senegal (national mean of 'flagged HAZ' 3.4%).

## Discussion

This study provides a quantification of household survey data quality at a 5 × 5 km spatial resolution in 35 countries in Africa for household surveys conducted between 2006 and 2022. Geocoded household survey data are increasingly used by researchers and practitioners. They also serve to track the achievement of a wide range of the United Nations (UN) development goals[72]. Identifying subnational localities with poor data quality could enable household survey data users to leverage additional tools and data when operating in these regions or when making causal inferences related to them. It also contributes to

ongoing efforts by the DHS program to improve data quality in these areas, such as reducing non-response for example by providing local survey teams with better training or more resources[37].

Our estimates highlighted a striking variation in data quality at district and local levels, which so far have been masked in cross-country or cross-survey studies. We also found that the prevalence of data errors by survey location was systematically increasing with distance from settlements, resulting in worse data quality for populations that live in remote rural areas. Systematically distributed measurement errors have potentially major negative implications for research, particularly if the direction and the magnitude of the bias are unknown. The spatial bias did not substantially change between surveys for most of the included countries with multiple surveys, which indicates a potentially persistent problem beyond potential one-off effects of single survey rounds. Figure 3 identifies geographical areas where high statistical uncertainty overlaps with systematic measurement and missing data, which together threaten the quality of the insights that can be drawn from household survey data. Across and within the 35 African countries, we found little correlation between standard deviations and systematic data errors, suggesting that these follow relatively independent spatial patterns that are obscured by looking at national measures alone.

The DHS program is considered to be a gold standard among periodic household surveys and is widely trusted. Survey experts or field enumerators who have studied measurement errors in DHS data have created awareness for the potential of systematic measurement errors and missing data using cross-country and cross-survey analysis to assess the variation in measurement errors, their magnitude, as well as potential factors and consequences[28,30,38,46]. However, the spatial dimension of systematic measurement errors at fine-grained local levels, where decisions and planning take place, has been unknown. There is a large community of data users who rely on household survey data for conducting research or planning health interventions, but who lack the specific knowledge of experts involved in enumeration. Unaware of the magnitude of this problem, scientists or funding agencies alike have so far assumed either sufficiently high survey data quality or a relatively innocuous, random distribution of errors. High-resolution mapping studies may or may not statistically account for randomly distributed measurement errors. Accounting for non-randomly distributed errors that this study reports will be difficult without additional data, because the direction and the magnitude of the biases are often unknowable[73]. For applied researchers in particular, the main implication is that data quality needs to be considered and addressed when analyzing data from remote populations.

Health and development efforts are typically implemented on community and district levels. Our estimates highlight districts and regions across Africa in which survey-based anthropometric and demographic information may be too uncertain or biased to support inference, conduct policymaking or inform intervention[74]. Even though researchers and funding bodies have a professional obligation to effectively communicate the uncertainty around estimates, data errors often fail to catch the attention of policymakers, government agents or fellow researchers. Mapping estimates of measurement errors can improve the awareness of these users in their work with underlying anthropometric, health, and demographic data. Identifying data quality issues on national levels alone may not be adequate, because data quality varies by location (Fig. 2) as do standard deviations, and there is little apparent correlation between them (Fig. 3).

Our findings corroborate and expand the details needed to understand data quality challenges of the "last mile" problem in global public health and development more broadly[75]. Many researchers work with these data without knowing how large the variation in data quality really is, and how data quality issues can include uncertainty as well as non-random missing data which varies by remoteness. Our analysis is based on a twofold empirical approach. For one, we directly document the increase in the prevalence of errors for sampled areas in

more remote locations using raw DHS data. Because the remoteness penalty varies across countries and across DHS variables, with HAZ being the least affected, DHS data users studying remote populations are advised to assess, on a case-by-case basis, the potential for systematic missingness and measurement error to affect the conclusions they wish to draw form the data. For another, we predict data errors for the non-sampled locations to better illustrate the extent of data quality challenges and to compare statistical uncertainty with low data quality which are two distinct problems. With better access to information about data quality, researchers will be better equipped to use the data and address any limitations of the data if they can do so, or else at least adjust their data interpretations as needed.

Our modeling framework is, like any statistical model, an abstraction of reality and subject to several limiting simplifications. Spatial data gaps between DHS survey locations may introduce varying degrees of uncertainty in our estimates. These considerations also apply – to a lesser extent – to the aggregated analyses and are relevant to any geolocated household-survey based data set. Also, survey locations are randomly displaced in space for data confidentiality reasons, further increasing the level of spatial uncertainty. However, the covariate layers we use to extract external information for each DHS cluster do not vary substantially at this high resolution and we expect that our overall findings will remain essentially unchanged. Another potentially limiting factor is the volatile and unpredictable nature of armed conflict and local unrest, population movements, and the impacts of local political instability on the work of survey teams. Conflicts limit the reliability of coverage from household-based surveys due to an inability to sample in unsafe and insecure areas[17,18]. Further limitations include that we choose one model specification out of many, which holds potential influence on the predictions. Our estimates include uncertainty which we do not show, as the maps are merely point estimates.

Our intent is to illustrate the magnitude of this problem, not discuss underlying causes, even though our measures, such as nighttime light emissions, include an economic and an accessibility dimension. Following recent studies that have revealed considerable heterogeneity in important health and nutrition indicators of human well-being at the subnational level, which had previously been masked by country-wide measures[10,76,77], we hope to motivate more research into the causes of the subnational heterogeneity in data quality, including low birth registration, low access to education[47], migration[78], or high levels of poverty[79]. Remote communities, which are more likely to suffer from these problems, may also be more difficult for enumerators to work in. Local languages may differ from those native to enumerators or travel to remote communities is relatively more demanding and resource intensive. Error type-specific interventions and more research will be needed to sufficiently address these concerns. In practice, the drawbacks of reforms in survey design and fieldwork practices such as increased financial burden will need to be weighed against the benefits on a case-by-case basis. Given that many health practitioners and governments use DHS data directly to inform policy and interventions, improving understanding of data quality within local contexts is essential for human development. Our estimates of the magnitudes of the errors will allow governments and agencies to plan for the next rounds of data collection with better targeted resources.

## Methods
### DHS data
We compiled a database of publicly available geocoded DHS data for 35 African countries. DHS data are best suited for the scope of the study, because they are publicly available, geocoded, and more comprehensive in time and space than data from other large survey programs such as MICS or LSMS[6,80,81]. To demonstrate the deteriorating data quality in remote areas we directly calculated the prevalence of data errors in the most recent DHS survey round to avoid reporting a purely artificial, mechanical relationship arising from regressing

predictions generated by a geostatistical model, which are a function of distance to nightlight, on distance to nightlight. To illustrate within-country variation in data quality as well as the relationship between data errors and sampling uncertainty, we moved beyond the DHS survey locations and employed Bayesian model-based geostatistics, using covariates from gridded high-resolution datasets to predict data quality beyond surveyed areas. The geostatistical model (see below) produced 5 × 5 km gridded estimates (roughly equivalent to villages in rural areas) of three widely used data quality indicators, which we aggregated to district ('admin-2') levels and national levels using population weights. The number of DHS survey rounds varied across African countries. In 24 countries with more than one survey round since 2006, we used the most recent survey (Supplementary Table 10). We extracted relevant variables from the DHS children file, from the individual women's file and the household roster file. In DHS data, the individual and household survey questionnaires are linked to a survey location (termed 'cluster' or 'primary sampling unit') that represents a set of neighboring households or a village structure. The data we derived from the most recent surveys came from 20,077 DHS clusters. For most clusters, geolocation information in the form of GPS coordinates were available. For the data from the Republic of the Congo (equivalent to 1.6% of the sample size), no geoinformation were available. To impute likely locations for these primary sampling units, we sampled 1000 candidate locations for each 'latent' survey location, based on population density rasters for each sub-national unit. The resulting candidate positions were then clustered using a k-means procedure and the centroids of the resulting clusters were used as pseudo-location. For each of these pseudo-locations, the sub-national average value of the indicator of interest was assigned. This procedure closely followed the imputation strategy that has been used for similar problems in recent literature[10].

## Covariate data

The modeling framework includes the following geospatial covariates which we constructed from freely available gridded datasets: population density (1), malaria incidence (2), terrain ruggedness (3), nightlight intensity (NTL) and settlement data (4). We believe a larger set of covariates would add no substantial explanatory power or change the argument of the study. For all covariates, data from the closest grid cell and matching the respective survey year were used for estimation. Where these were unavailable, data from the year closest to the respective survey year were included in the modeling framework. For prediction, covariates closest to the year 2013 were used. Prior to introducing the variables in our statistical modeling framework, standard temporal and spatial pre-processing steps have been conducted (Supplementary Tables 1 and 2).

Spatial processing. Both malaria incidence and population density are available at a resolution finer than the resolution of the nightlight data. To align the resolution covariate rasters, the input covariates 'ruggedness', 'malaria incidence' and 'population density' were resampled from their native resolution to match the native resolution of the nightlight intensity rasters (roughly 5 × 5 km at the equator line). Bilinear interpolation was used as resampling technique in all cases.

Administrative boundaries: We used GADM shape files in version 4.1 to define the national and sub-national boundaries used for spatial analysis[82]. In general, we used the smallest subdivisions available, which were equivalent to admin-2 for most countries and to admin-1 for Lesotho.

Gridded population density: We obtained the gridded data on population density from the Gridded Population of the World (GPW) database, version 4[83]. The data are available in steps of five years. In general, we selected the year closest to the relevant year (e.g., survey year or year of predictions) for the analysis.

Malaria Incidence: We used a standard data set on malaria incidence, obtained from the Malaria Atlas Project (MAP)[19]. Specifically, we

used data on the incidence of *Plasmodium falciparum*. We included the closest available year to the respective survey round in the analysis. The year 2013 was used for predictions.

Ruggedness: Terrain ruggedness (3) was extracted from a data set assembled and collected by Nathan Nunn and Diego Puga in 2012. The raw data are available from the personal website of Diego Puga[84]. The data come in form of a 30 arc-second grid covering the globe. We assumed that terrain ruggedness is time-invariant.

Nighttime light emissions: Satellite images recording nighttime light (NTL) emissions are frequently used to estimate economic activity[11]. We used the temporally NTL dataset generated by Li, Zhou, Zhao & Zhao (2020) which harmonizes inter-calibrated NTL data from the Defense Meteorological Satellite Program Defense Operational Line Scanner (DMSP-OLS) and simulated DMSP-OLS like NTL observations from the newer Visible Infrared Imaging Radiometer (VIIRS) data[85]. DMSP-OLS data are composite images in which each 30 arc second pixel (about 1 km at the equator) gives the annual average brightness level in units of digital numbers (DN) ranging from 0 (no light) to 63 (maximum reported light). For model estimation, we used the corrected DMSP NTL data covering observations from 1992 to 2013 across different satellites (F10, F12, F14, F15, F16, and F18), falling in the closest year that was available to the respective year closest to each survey year. For prediction, we used the DMSP NTL data for the year 2013. The improved data provided by Li and co-authors are temporally consistent correcting for varied atmospheric conditions, satellite shift or sensor degradation. In the analysis our 'distance to nightlight' variable measures the distance from the nearest NTL emissions of DN 15. We built binomial regression models with logistic link, regressing the respective data quality indicators computed from raw DHS data on distance to nightlight and fixed effects. We used raw DHS data to avoid reporting a purely artificial, mechanical relationship arising from regressing the predictions of data quality generated by a geostatistical model. Across 35 African countries, the average distance to the nearest settlement emitting NTL of DN 15 or higher was 64.8 km and 90% of clusters were located within 163.1 km from a settlement of DN 15 or higher. Thresholds delineating rural from urban areas vary across space and time[66]. Large rural settlements may remain dark on NTL maps for lack of access to electricity[86]. NTL emissions of DN 15, our main threshold, lie well above the lowest identified NTL thresholds of about DN 5 emitted from villages in Senegal and Mali with more than 40 streetlights[67]. DN 15 is below DN 20 assumed for urban, built environments such as in smaller cities[66]. In line with this literature, we concluded that in most of sub-Saharan Africa thresholds above DN 15 very likely include settlements with public streetlights and electrified buildings indicating a minimum degree of urban infrastructure and economic activity[65].

Settlement data: We replaced NTL emissions with other measures of human settlements to see if our findings remained robust. The bias in the distribution of the errors remained apparent with auxiliary a) high-resolution settlement data and b) data on urban African agglomerations. Satellite-based high-resolution settlement data were taken from Marconcini et al.[68]. These data do not distinguish in the geographic extension of settlements and include tiny hamlets and suburbs. The average distance to the nearest settlement was 2.5 km. 90% of clusters were located within 5.1 km from a settlement. The deterioration of data quality was apparent in most sub-Saharan African countries even for these short distances. In contrast, Africapolis data collect larger urban agglomerations of at least 10,000 inhabitants[87]. The average distance to the nearest urban agglomeration of at least 10,000 inhabitants was 61.6 km. 90% of clusters were located within a distance of 167.1 km. Data from the Africapolis dataset are partly constructed from census data or population records which may be outdated or of varying reliability.

Covariate analysis: All covariates were examined for multi-collinearity in a non-spatial modelling framework for each indicator.

We calculated the variance inflation factor (VIF) for each covariate in each case and concluded that there was no evidence of multicollinearity when the VIF value is <4.0. This was done for each modelling region, namely Central Africa (CA), East Africa (EA), Southern Africa (SA) and West Africa (WA). Where covariate information was missing at the cluster and grid level, we imputed using kriging interpolation based on covariate information from nearby locations. This was done for Lesotho, for which malaria prevalence data are not available. Both cluster and grid level data for this country were imputed using cluster level data from neighboring countries.

## Geostatistical model

Each of our three data quality indicators was modelled in each of the four geographic regions, and the outputs of the regional models were combined to form a continuous map for each indicator. We describe each indicator in detail in the Supplementary Note 2. Countries were assigned to one of the four regions (Supplementary Fig. 16). This sped up computation and leverages regional relationships between the indicators and the geospatial covariates. Our INLA-SPDE approach, which we explain in the exploratory data analysis, requires the specification of a fine triangulation mesh to approximate the spatial random effect. An example of a spherical mesh for West Africa is shown in Supplementary Fig. 1.

## Exploratory data analysis

To examine the relationships between the covariates and each of the indicators, we used the empirical logit transformation of the indicators given by Eq. (1):

$$y^*(\mathbf{s_i}) = \log\left(\frac{y(\mathbf{s_i}) + 0.5}{m(\mathbf{s_i}) - y(\mathbf{s_i}) + 0.5}\right), \; i = 1, \; \ldots, n, \tag{1}$$

where $y(\mathbf{s_i})$ denotes the number of individuals possessing the attribute being modelled at spatial location $\mathbf{s_i}$ (represented using the longitude and latitude coordinates) and $m(\mathbf{s_i})$ is the number of individuals sampled from that location. For example, for 'incomplete age', $y(\mathbf{s_i})$ is the number of individuals with missing age out of $m(\mathbf{s_i})$ individuals sampled from location $\mathbf{s_i}$. To improve the linear associations between the covariates and the logit-transformed indicators as is assumed in a binomial regression context, we log-transform the covariates, considering that these have heavily skewed distributions. We fit non-spatial binomial regression models and compute the Variance Inflation Factor (VIF) to check for (multi)collinearity[88]. A VIF value greater than 4.0 is indicative of the presence of (multi)collinearity among the covariates. Also, using the residuals from the non-spatial models, we calculate the empirical variograms of the indicators to check for the presence of residual spatial correlation. Assuming the presence of considerable spatial correlation in the residuals, we specified a spatial model for the analysis, described in the next section. All the covariates were standardized to have a mean of zero and variance of one prior to model-fitting.

## Geostatistical model, model-fitting and prediction

To predict the data quality indicators at 5 × 5 km resolution and the district level, we fitted a Bayesian geostatistical model with a binomial likelihood. Using similar notation as before, the first level of the model can be expressed as Eq. (2):

$$Y(\mathbf{s_i})|m(\mathbf{s_i}) \sim \text{Binomial}(m(\mathbf{s_i}), p(\mathbf{s_i})) \tag{2}$$

where $p(\mathbf{s_i})$ is the underlying true proportion of individuals possessing the attribute being modelled, e.g., incomplete age, at location $\mathbf{s_i}$. We model $p(\mathbf{s_i})$ using the logistic regression model expressed as Eq. (3):

$$\text{logit}(p(\mathbf{s_i})) = \mathbf{x}(\mathbf{s_i})^T\boldsymbol{\beta} + \omega(\mathbf{s_i}) + \epsilon(\mathbf{s_i}) \tag{3}$$

where $\mathbf{x}(\mathbf{s_i})$ is a vector of covariate data associated with location $\mathbf{s_i}$, $\boldsymbol{\beta}$ are the corresponding regression coefficients, $\epsilon(\mathbf{s_i})$ is an independent and identically distributed (iid) Gaussian random effect with variance $\sigma_\epsilon^2$ used to model non-spatial residual variation, and $\omega(\mathbf{s_i})$ is a Gaussian spatial random effect used to capture residual spatial correlation in the model; i.e. $\omega = (\omega(\mathbf{s_1}), \ldots, \omega(\mathbf{s_n}))^T \sim N(0, \boldsymbol{\Sigma_\omega})$. $\boldsymbol{\Sigma_\omega}$ is assumed to follow the Matérn covariance function[24] given by

$$\sum_{\boldsymbol{\omega}}\left(\mathbf{s_i}, \mathbf{s_j}\right) = \frac{\sigma^2}{2^{\nu-1}\Gamma(\nu)}(\kappa||\mathbf{s_i} - \mathbf{s_j}||)^\nu K_\nu(\kappa||\mathbf{s_i} - \mathbf{s_j}||) \tag{4}$$

where $|| \cdot ||$ denotes the distance between cluster locations $\mathbf{s_i}$ and $\mathbf{s_j}$, $\sigma^2$ is the marginal variance of the spatial process, $\kappa$ is a scaling parameter related to the range $r(r = \frac{\sqrt{8\nu}}{\kappa})$ – the distance at which spatial correlation is close to 0.1, and $K_\nu$ is the modified Bessel function of the second kind and order $\nu > 0$. Further, for identifiability reasons, we set $\nu = 1$[89]. We fit model (1) for each region (Supplementary Fig. 16). A Bayesian approach is adopted for the analysis, which is implemented using the integrated nested Laplace approximation – stochastic partial differential equation (INLA-SPDE) approach[89,90]. To complete the model specification, we place a non-informative $N(0, 10^3\mathbf{I})$ prior on the regression coefficients, $\boldsymbol{\beta}$. A penalized complexity (PC) prior introduced in Simpson et al. 2017 is set on $\sigma_\epsilon$ such that $p(\sigma_\epsilon > 3) = 0.01$[91,92]. Similarly, following Fuglstad et al. 2018, a joint PC prior is placed on the covariance parameters of the spatial random effect, $\omega$[93]. These are: $p(r < r_0) = 0.01$ and $p(\sigma > 3) = 0.01$, with $r_0$ chosen to be 5% of the size of the region. The SPDE approach involves a triangulation of the spatial domain to approximate $\omega$. A spherical mesh was constructed for this approximation using the boundary points and the data locations falling within each region. The maximum triangle edge length is set to 15 km in the inner mesh and 100 km (50 km in Southern Africa) in the outer mesh, with an offset of 500 km in each case. With the spherical mesh, the distance matrix needed to fit the Matérn covariance function is calculated as the great circle distance along the surface of the Earth. To implement this in practice, the data locations and boundary points for each region are first converted to three-dimensional coordinates on the unit sphere. These are then used to construct a mesh on a spherical manifold, $S^2$, on which the SPDE is defined. From each of the fitted models, we generate 1000 samples from the posterior distributions of the parameters of the model, as well as from the posterior predictive distributions of the indicators for each of the prediction locations, i.e., the 5 × 5-km grid cells. The latter are then used to calculate the district level estimates as population-weighted averages taken over all the grid cells falling within each district.

## Model validation

To assess the performance of the fitted models for out-of-sample prediction, we adopted k-fold cross-validation, setting k = 10. The cross-validation folds were created as random splits of all the data locations falling within each region. The following model evaluation metrics were computed using the observed ($p(\mathbf{s_i})$) and predicted values ($\hat{p}(\mathbf{s_i})$) of the indicators (on the probability scale) for $i = 1, \ldots, m$ validation locations and averaged over the 10 subsets:

$$\text{Percentage bias } (\%Bias = 100^* \sum_{i=1}^{m}(\hat{p}(\mathbf{s_i}) - p(\mathbf{s_i})) / \sum_{i=1}^{m}(p(\mathbf{s_i})) \tag{5}$$

$$\text{root mean square error } (RMSE = \sqrt{\sum_i(\hat{p}(\mathbf{s_i}) - p(\mathbf{s_i}))^2/m} \tag{6}$$

$$\text{mean absolute error } (MAE = \sum_i|\hat{p}(\mathbf{s_i}) - p(\mathbf{s_i})|/m) \tag{7}$$

and the Pearson correlation coefficient. The closer the values of % Bias, RMSE and MAE are to zero, the better the predictions. Correlation values close to one indicate better predictive ability.

We found no substantial non-linear associations between the covariates and the indicators in each region (Supplementary Fig. 17 for 'age heaping'). There was also no evidence of (multi)collinearity in the data as the VIF values of the covariates are estimated below 4.0 in each case. For all the indicators, we found evidence of significant residual spatial correlation in the non-spatial regression models fitted using all four covariates (Supplementary Fig. 18), justifying the use of geostatistical models for prediction.

Predictive performance: Our primary goal was to provide predictions of data quality indicators and associated uncertainties across sub-Sahara Africa at a high resolution. This means that we focused on the predictive performance of the fitted models rather than causal interpretability. Based on the correlations between observed and predicted values (Supplementary Table 3) the best predictive performance was observed in Central Africa, where the estimated correlations were greater than 0.58 for all the indicators (except 'flagged HAZ') and the worst predictive performance was observed in Southern Africa (SA). Other validation metrics indicate that the fitted models produced reasonably accurate predictions. The RMSE values were ≤0.18 in most cases while the MAE were ≤0.15 in most cases. The maximum RMSE was 0.24 ('stunting prevalence' - SA and 'incomplete age' - WA), the maximum MAE was 0.20 ('stunting prevalence'; SA) and the maximum % Bias (in absolute value) was 2.97 ('stunting prevalence'; SA).

Interpretation of Estimated Parameters: The covariates were more frequently significant predictors of 'incomplete age', 'stunting prevalence' and 'flagged HAZ', compared to other indicators and were hence most informative for these indicators (Supplementary Tables 4–8). In terms of the relative magnitudes of the estimated regression coefficients, no covariate uniformly has the largest coefficient (in absolute value) in all the regions for any of the indicators. We note that while most of the estimated relationships between the covariates and the outcome indicators are intuitively in the expected direction, others are not. There are a number of reasons for this, including undetected collinearity or measurement error in the covariates. However, as noted earlier, this is not a concern as this study focuses on prediction. In all cases the estimates of the spatial range ($\hat{r}$), the spatial variance ($\hat{\sigma}^2$) and the iid error variance ($\hat{\sigma}^2_\epsilon$) were all significant, confirming the contributions of the spatial term $\omega$ and the non-spatial errors $\epsilon(\mathbf{s_1}), \ldots, \epsilon(\mathbf{s_n})$ to explaining residual variation in the fitted models.

### Reporting summary
Further information on research design is available in the Nature Portfolio Reporting Summary linked to this article.

## Data availability
The findings of this study are supported by DHS datasets that are publicly available online at https://dhsprogram.com/. A detailed description of the selected DHS datasets and completed DHS rounds per country can be found in Supplementary Table 9. Other referenced data used for covariates are taken from freely available gridded datasets. (1) The Gridded Population of the World, Version 4 (GPWv4): Population Count. https://doi.org/10.7927/H4X63JVC. (2) The Malaria Atlas Project (MAP) https://data.malariaatlas.org/. (3) Terrain ruggedness was extracted from a data set assembled and collected by Nathan Nunn and Diego Puga in 2012 https://diegopuga.org/data/rugged/. (4) The nighttime light emissions dataset was generated by Li, Zhou, Zhao & Zhao https://doi.org/10.6084/m9.figshare.9828827.v2. (5) Settlements in the study were delineated by using both the satellite-based high-resolution settlement data from Marconcini et al. https://doi.org/10.6084/m9.figshare.c.4712852 and the Africapolis dataset at https://

africapolis.org/en/data. The primary generated in this study and the data to replicate all findings are available at https://data.worldpop.org/repo/prj/dhs/SSA/data_quality.zip.

## Code availability
All computer code used in the analyses is available at https://doi.org/10.5281/zenodo.14892010. All maps and figures in the manuscript were generated by the authors using R v. 4.1.1 and ArcGIS Desktop v10.6.

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

## Acknowledgements

V.S. acknowledges the support of the Austrian National Bank (Anniversary Fund) Grant No. 18157 and the Austrian Science Fund (FWF). This research was funded in whole, or in part, by the Austrian Science Fund (FWF) [P 34329-G]. For the purpose of open access, the author has applied a CC BY public copyright license to any Author Accepted Manuscript version arising from this submission. PW acknowledges the support of Feed the Future Food Systems for Innovation Lab, funded by the United States Agency for International Development, Cooperative Agreement No. 7200AA21LE00001. Funding sources had no role in the development of this study, the writing, or the decision to submit for publication.

## Author contributions

V.S., G.Z., A.B.F., S.L., S.E.K.B., and P.W. conceived and planned the study. E.C.U. and G.Z. obtained, extracted, processed, and geopositioned the data. A.S. developed and wrote the literature review. E.C.U. and G.Z. directly accessed and verified the underlying data reported in the manuscript. S.L. compiled the geodata and wrote the computer code for the visualizations. E.C.U. and G.Z. constructed covariate data layers. E.C.U. and G.Z. wrote the computer code and designed and carried out the statistical analyses with input from A.B.F., A.J.T., S.L., and V.S. A.W.S., E.C.U., G.Z., S.L., and V.S. prepared tables and figures. A.J.T., S.E.K.B., and P.W. provided intellectual inputs into all aspects of this study. M.B. built the online data visualization tool. V.S., A.S., and A.B.F. wrote the first draft of the manuscript, and V.S., E.C.U., A.B.F., G.Z., S.L., S.E.K.B., A.S., M.B., A.J.T., and P.W. contributed to subsequent revisions.

## Competing interests

The authors declare no competing interests.
