## [Transparent Peer Review file · Nature Communications]

Subnational variations in the quality of household survey data in sub-Saharan Africa

Corresponding Author: Dr Valentin Seidler

Version 0:

Reviewer comments:

Reviewer #1

(Remarks to the Author)

This paper addresses a perennial concern of all scientists working on human development in Sub-Saharan Africa (SSA): the quality of data. A decade ago, Devarajan (2013) argued that SSA faced a "statistical tragedy" in that "GDP accounts use old methods, population censuses are out of date, and poverty estimates are infrequent and often not comparable over time. The proximate reasons have to do with weak capacity, inadequate funding, and a lack of coordination of statistical activities." See also the extensive discussion in Jerven (2013).

Many of these shortcomings remain, although there has been progress. For instance, over the past decade the scope of micro-level data has expanded in terms of country coverage, in terms of time coverage, and in terms of spatial coverage, including the geolocation of entities surveyed.

Examples that are now regularly used by researchers include the Living Standards Measurement Study (LSMS) surveys (LSMS-ISA includes Agriculture), Demographic and Health Surveys (DHS), Household Income and Expenditure Surveys (HIES), Labor Force Surveys (LFS) and the AfroGrid data which now cover 1989-2020 and provide socio-economic, climatic and conflict data at the .5 x .5 degree grid cell level. In addition, satellite data has become available, and many countries implemented reforms within their statistical agencies, and updated/ recalibrated their basic economic data (e.g. Ghana, South Africa).

Still, many shortcomings remain, and amongst these are measurement errors. Development economists have done quite some work on these, including on the sub-national level. It is well known in development policy circles that data in remote areas are less reliable and that this contribute to the potential greater vulnerability of populations in such remoter areas - indeed the concepts of data gaps, data divides, data justice and data inequality are well explored (e.g. Adebisi and Lucero-Priso, 2022).

This is mentioned as it seems that the current paper seems unaware of this work and tackles the issue of measurement errors only from a health / demographic perspective (illness and malnutrition), and seems to suggest that policy makers are unaware of data gaps and data divides. As such, it misses much that has been learned in development economics. Back in 2009 Naudé et al (2009) for instance argued that the error term in a regression of sub-national income levels on Human Development Indicators in South Africa reflects unexplained variations in human development that carries potentially important information for sub-national policy interventions.

More recent examples of work on measurement errors in micro-level surveys in SSA in this regard includes : Abay (2020), Carletto et al (2017), Carletto et al (2013), Desiere & Jolliffe (2018), Gollin and Udry (2021), Gibson and Kim (2010), Hyslop, and Imbens (2001). This is just a modest selection. Much of the studies cited here has a been concerned with understanding and correcting for classical and nonclassical measurement errors in agricultural surveys (e.g. LSMS-ISA) - particularly relevant given the importance of agriculture as livelihood across the spatial economy of SSA.

Brück et al (2022) constructed a spatial fragility exposure index using Kenyan data, calling for taking into account spatial heterogeneity in exposure to conflict to the reliability of data. This is mentioned since the current paper also ignores conflict, and claims that "The modeling framework includes the following geospatial covariates which we constructed from freely available gridded datasets: population density (1), malaria incidence (2), terrain ruggedness (3), nightlight intensity (NTL)

and settlement data (4). We believe a larger set of covariates would add no substantial explanatory power or change the argument of the study." Certainly, many authors working on conflict in SSA, the nature of vulnerability and fragility on the continent, and the impact of conflict on health, poverty, and migration, would strongly argue that the authors miss a very important spatial determinant of outcomes, but moreover a variable that affects the quality of data collected - in obvious ways (and perhaps this is even more important in Central Africa, where the authors found the best predictive performance of their model....)

Hence, it is not entirely accurate as the authors claim that "we still lack knowledge about the spatial distribution and magnitudes of data errors where it matters most - on subnational and district levels." The studies cited have resulted in good knowledge about measurement errors in agriculture, the role of conflict in measurement errors, etc.

Having mentioned this blindspot however, it has to be emphasised that the current paper does make an interesting contribution to this growing and evolving debate. Their core arguments - that the spatial dimension of policy making requires good sub-national and regional data, and that the reliability of data across sub-national regions are not always equally reliable - are worth stressing. Moreover, their methodology is both creative and rigorous in terms of trying to quantify the measurement errors across sub-national regions in SSA.

The only methodological issues I would flag is the likely high correlation between conflict across space and measurement errors in their variables of interest as already mentioned. Related to this, is their lack of using longitudinal data, which has typically been a central strategy used by economists to reduce the impact of unobserved heterogeneity across time where these have been time invariant. For instance, some of the geo-variables that the current study control for, such as terrain ruggedness, are time invariant; to the extent that it would affect nutritional outcomes, using longitudinal data would control for this. The changes in measurement errors over time is therefore a non trivial issue and somehow not discussed.

Finally, given that it is already well known that data divides and gaps exists, that remote populations are more vulnerable as a result, and that work on measurements errors in surveys are growing and ongoing, the questions that the current paper do not fully answer are "so what? what should researchers and policy makers be aware of? Can their methods shed light on how to reduce measurement errors?" The policy and research implications should be highlighted, including being stated upfront. Otherwise the reader will come away from the paper impressed by the data analytics, but with the feeling that not much has been found that was not already known or can be reasonably assumed.

REFERENCES:

Abay, K.A. (2020). Measurement errors in agricultural data and their implications on marginal returns to modern agricultural inputs, *Agricultural Economics*, 51 (3): 323-341; <https://doi.org/10.1111/agec.12557>

Adebisi YA, Lucero-Prisno DE 3rd. (2022). Fixing Data Gaps for Population Health in Africa: An Urgent Need. *Int J Public Health*. Nov 10;67:1605418. doi: 10.3389/ijph.2022.1605418. PMID: 36439279; PMCID: PMC9684304.

Carletto, G., Gourlay, S., Murray, S., & Zezza, A. (2017). Cheaper, faster, and more than good enough: Is GPS the new gold standard in land area measurement? *Survey Research Methods*, 11(3), 235–265.

Carletto, C., Savastano, S., & Zezza, A. (2013). Fact or artefact: The impact of measurement errors on the farm size–productivity relationship. *Journal of Development Economics*, 103, 254–261.

Desiere, S., & Jolliffe, D. (2018). Land productivity and plot size: Is measurement error driving the inverse relationship? *Journal of Development Economics*, 130(1), 84–98

Devarajan, Shantayanan. "Africa's statistical tragedy." *Review of Income and Wealth* 59 (2013): S9-S15.

Gollin, D. and Udry, C. (2021). Heterogeneity, Measurement Error, and Misallocation: Evidence from African Agriculture, *Journal of Political Economy*, 129 (1).

Gibson, J., & Kim, B. (2010). Non-classical measurement error in long-term retrospective surveys. *Oxford Bulletin of Economics and Statistics*, 72, 687–695.

Hyslop, D. R., & Imbens, G. W. (2001). Bias from classical and other forms of measurement error. *Journal of Business & Economic Statistics*, 19(4), 475–481.

Jerven, M. (2013). *Poor numbers: How we are misled by African development statistics and what to do about it*. Ithaca, NY: Cornell University Press.

Naude, W., McGillivray, M. and Rossouw, S. (2009). Measuring the Vulnerability of Subnational Regions in South Africa, *Oxford Development Studies*, 37:3, 249-276, DOI: 10.1080/13600810903085800

Reviewer #2

(Remarks to the Author)

Subnational variations in the quality of household survey data in sub-Saharan Africa

Summary:

In general, the manuscript presents a comprehensive examination of the quality of survey data, particularly DHS surveys, shedding light on heterogeneity in subnational distribution of sampling and measurement errors. The authors employed a systematic and rigorous approach to create awareness on the quality and utility of survey data for subnational level estimation, particularly at lower level of administration units, drawing upon a rich body of literature and empirical evidence from 33 countries in sub-Saharan Africa. While the manuscript contributes valuable insights to the field of survey research to generate policy- and program-related evidence, there are several areas that require attention and refinement.

Strengths:

Comprehensive Literature Review: The manuscript provides a thorough review of existing literature on surveys related to their quality and utility in low and middle-income countries. This review helps readers contextualize the research within the broader domain of survey methodology and serves as a solid foundation for further utility of these data at subnational level.

Methodological Rigor: The manuscript employs a methodologically sound approach to assess quality of survey data to perform subnational level analysis. The authors discuss various quality indicators: (1) incomplete age, (2) age heaping, and (3) 'flagged HAZ', providing a holistic view of the challenges faced in subnational level analysis of survey data.

Empirical Evidence: The inclusion of empirical data from about 33 countries in sub-Saharan countries strengthens the manuscript's credibility and practical relevance. Such exhaustively analyzed data provide valuable insights into the complexities and nuances of survey data quality issues when it comes to subnational level analysis.

Incorporation of visual aids: Visual aids, such as tables, charts, and graphs, could be employed to illustrate key findings and trends. Visual representations can help convey complex information more effectively and make the manuscript more engaging. The authors did a great job on this.

Recommendations: The manuscript offers practical recommendations for researchers to better understand the causes of heterogeneity in data quality (such as lack of birth registrations, lower access to education, migration, or poverty). The manuscript highlights that remote communities are likely the ones who are missed out from such surveys and hence are being misrepresented at subnational level of analysis. The paper recommends that local contexts need to be considered when using survey data to inform policies and interventions, particularly in lower level of administrative units. The authors of the manuscript also made available their estimates of the magnitude of errors, which will be of significant value to researchers, health practitioners and agencies in planning future surveys. These recommendations are well-grounded in the analysis and synthesis of existing literature and survey data from 33 countries.

Areas for Improvement:

Clarity of Presentation: The manuscript could benefit from improved clarity in its presentation of analysis results on the distribution of errors. Some paragraphs are densely packed with information, making it challenging for readers to follow the arguments and findings. Clearer organization (such as using tables) and more concise language would enhance readability.

Methodological transparency: While the manuscript discusses various quality indicators, there is room for greater transparency in the methodology used for data analysis and the justification in selecting the type of survey (DHS over MICS), geography (Sub-Saharan over all countries worldwide), year of survey, and subnational level of analysis (level-2 admin over level-3 and level-4 admin for some countries). Providing more details on the criteria used for assessing data quality from these perspectives would enhance the manuscript's rigor.

Given DHS surveys have become more robust in terms of sample size and methodology over time, it is not clear why authors chose to limit their analysis to surveys between 2006-2016 (for some countries goes to 2018/19).

Authors indicated that, in countries with more than one survey round since 2006, the most recent survey entered the study (completed DHS rounds and selected datasets in Supplementary Table S9). However, this is not the case for some countries. For example, the following countries have more recent data than the one described in table S9, and I recommend that authors consider exploring these datasets.

- Burkina Faso – 2021
- Cote D'Ivoire - 2021
- Ethiopia – 2019
- Gabon – 2019/21
- Ghana – 2019
- Guinea – 2021
- Kenya – 2022
- Liberia – 2019/20
- Madagascar – 2021
- Mauritania – 2019/21
- Rwanda – 2019/20

For the results to be more informative, as the study suggested, countries with recent DHS survey data with geographic details and perhaps with more sample size and greater reach to remote areas need to be incorporated in the analysis.

Addressing limitations: The manuscript should acknowledge its own limitations and potential sources of bias. An honest appraisal of the study's limitations would enhance the manuscript's credibility and assist readers in interpreting the findings.

Practical applications: While the manuscript offers recommendations for improving survey data quality in future, it would be valuable to provide concrete examples of how these recommendations can be applied in real research scenarios. This would assist readers in translating theory into practice. For instance, most of these countries have started collecting routine facility data, and therefore, it would be more helpful to expand the discussion on how to leverage such datasets as appropriate, particularly for policy, planning and programming purposes.

Furthermore, it is likely that many development-related factors (such as increased access to roads, education, and telecommunication, etc.) brought opportunities to increase the geographic reach of DHS sampling. Given most countries have multiple surveys over the past two decades, it would be worthwhile to expand the discussion using results from Figure S9 to highlight whether or DHS sampling strategies took advantage of increased access to remote areas.

Overall, this manuscript makes a valuable contribution to the field of survey research by offering a comprehensive analysis of survey data quality at subnational level. With some revisions to improve clarity, transparency, and the incorporation of more recent datasets, it has the potential to become an even more valuable resource for researchers and practitioners. Additionally, addressing the limitations of the study and enhancing the practical applications of the recommendations would further strengthen the manuscript's impact.

Reviewer #3

(Remarks to the Author)

Report on "Subnational variations in the quality of household survey data in sub-Saharan Africa"

The paper analyzes unambiguous errors in the microdata of Demographic and Health Surveys from 33 African countries. The authors use measures of age heaping, biologically impossible height-for-age Z-scores, and missing data as their error indicators. They show substantial subnational variation in these errors, which is increasing with remoteness, but imperfectly correctly with traditional sampling uncertainty.

The basic premises of the paper strike me as valid, and a worthy motivation for the paper: in many low- and lower-middle income countries, DHS takes the place of administrative data as the gold standard for vital statistics. Furthermore, researchers (including myself) typically ignore subnational variation in data reliability.

I'll divide my comments between the three main results in the paper.

Result 1: Subnational variability of data quality

To my mind, this result really consists of two steps, and I'm a bit skeptical of the second:

Measuring errors in geocoded DHS data.

Extrapolating those errors down to a level of geographic specificity that the DHS does not (and is arguably not designed to) provide.

My doubts here are less with the method used to impute errors at, say, the district level, and more with the motivation and utility of doing so. The paper offers the following motivation:

"Assessments of household surveys have so far compared data quality across provinces or states (admin-1 level) or between rural and urban populations. This is unsatisfactory, because decisions are often made locally or on the level of districts (admin-2) and should ideally be based on reliable information from the same locality."

Yes, policy decisions happen on a sub-national basis, of course. But DHS data users are typically discouraged from producing or using estimates at the district level. The DHS program, as the authors know very well, typically only claims the surveys are representative at a higher administrative level, typically admin-1 or even higher.

It would be useful for the paper to illustrate the kinds of decisions currently being made on the basis of DHS data at the district level. At the risk of overstating my point, at present this section reads to me as "here's something you were never supposed to do with DHS data, and sure enough, it seems dubious to do so."

Finally, while the maps are useful, it would also be nice to have a clearer analytical statement of the "result" here. For instance, is the variation higher than would be produced by chance? I struggle to translate the finding into concrete words.

Result 2: Deteriorating data quality in remote areas

This section provides useful evidence that the errors documented above are not random, but systematically correlated with

observable geographic features.

The section refers to “measurement bias”, but I’m unclear exactly what bias this is. If analysts use the DHS to compute regional estimates of, say, child mortality, would the patterns of measurement error documented here bias the results somehow? Perhaps so, but it doesn’t seem obvious to me what kind of bias there would be. The paper would benefit from some analytical account of this, and/or an illustration.

(As an aside, in line with my comments on result 1, I can’t resist noting that the authors rely on raw DHS data, rather than their model imputation above, to study the relationship between remoteness and errors. This reinforces my doubts about the need for such imputations of error levels at high levels of geographic disaggregation!)

Result 3: Measurement errors and sampling uncertainty

I was surprised (in a good way) by the suggestion that measurement error and sampling uncertainty are not concentrated in the same geographic places. As with result 1, however, I didn’t quite see a clear statistical statement of this finding. Even a simple correlation coefficient would help (if this is reported and I missed it, apologies). Once again, the maps are an imperfect substitute for a clear, unambiguous statement or statistic that can be tested or compared to findings from elsewhere.

Lastly, while interesting for someone like me who often uses DHS data, the motivation for exploring this correlation was again not super clear to me. I am tempted to treat the (implied) low correlation between errors and uncertainty as good news, spreading the source of confusion more evenly across geographic space. But that’s a casual speculation, and I’d benefit from more concrete explanation of what we should conclude from this.

Version 1:

Reviewer comments:

Reviewer #1

(Remarks to the Author)

In my opinion they have adequately addressed these, and I am satisfied that the paper is suitable for publication.

(Remarks on code availability)

Reviewer #4

(Remarks to the Author)

Dear editor,

I am writing to peer review article number NCOMMS-23-43116A, titled “Subnational variations in the quality of household survey data in sub-Saharan Africa”.

This paper quantifies spatial variation in data quality in the most recent Demographic and Health Survey of 35 African countries, focusing on incomplete age, age heaping, and flagged HAZ as three examples of poor data quality. The paper then creates 5 km x 5 km surfaces of these indicators covering the 35 countries in order to help users of DHS data in academic and policy spaces know which regions of which countries likely have poorer data quality. Finally, the authors show that data quality worsens at longer distances from nightlights (using a DN 15 as the threshold), and they show that there is no correlation between their measures of data quality and measures of statistical uncertainty from sampling.

I will first offer more overarching comments and then offer more minor comments.

First, I enjoyed the paper and the focus on spatial variation in data quality in the DHS. I concur with the authors that the DHS has emerged as a very frequently used data source not only in academia but also in development and policy for within-country targeting of resources for health and development. Any novel work on the DHS that affects its use by this wide audience is therefore of important general interest. I therefore welcome the idea of quantifying measurement errors at subnational level for a large number of countries for users of the DHS.

My first question regards what follows the primary contribution. The paper could have gone in the direction of exploring what drives differences in data quality, as another referee has suggested, but this causal framework would call for different kinds of statistical analysis (exploiting longitudinal data, fixed effects, etc.). The authors have chosen not to go in that direction, which I don’t think is a problem, but it does mean that the remaining contributions of the paper should be substantive. The authors offer two add-ons to the surface mapping of measurement error: (a) they document that the problem becomes more severe as distance from urban areas (proxied by NTL) increases, and (b) the sampling variation is uncorrelated to the measurement error.

Regarding (a), as the authors already state, it is already known that “[DHS] data in remote areas can be less reliable

relative to urban areas" (line 51). If the result is not too novel, perhaps the authors can deepen the analysis by adding other variables predictive of measurement error. These could be household assets and characteristics, ethnicity and language, etc. Knowing whether other variables observable to DHS enumerators and DHS users are predictive of measurement error would be useful for targeting additional resources to improve data quality, or for analysts, to statistically account for lower data quality using other DHS variables.

Regarding (b), the authors could improve the motivation. On line 201 they explain that result 3 was explored because they were "curious to know to which extent high sampling uncertainty of stunting overlapped with the biologically implausible height-to-age measurements taken in the same population." A better motivation would explain why answering this question is important, and how the overlap (or lack thereof) is important for users of the DHS. High statistical uncertainty in rural places with small population (and sample) sizes is to be expected, and is in line with the DHS sampling strategy. So concluding that more sampling is needed there is counter to the DHS strategy. Moreover, investing in increased sample size and improved data quality might be at odds from a resources point of view. In summary, the documented correlations between uncertainty and measurement error, therefore, need to be better motivated for the reader.

A second question regards the authors' assertion of bias generated by the measurement error. For example on line 147 when discussing the second result, the authors argue that lower data quality in more rural areas generates a bias. Similarly, the paragraph beginning on line 265 discusses "non-randomly" distributed errors documented by the study. While this could be true if calculating a national estimate of population statistics, the relevant use case here is for localized statistics. For this purpose, it's unclear to me why there is necessarily bias, since missingness occurring randomly within the data distribution of rural observations (even if more frequent than more urban observations) would not generate a bias, only noisier estimates. Just because there is more missing data in rural areas does not mean it is non-random within the rural sample, which is what would affect bias in calculating population statistics for that region, or causal studies on household-level data. In fact, the authors cite a study by Finaret et al. [2018] which finds that bias from unusable HAZ data is likely small. This then begs the question: how important is this issue that the authors are documenting? Are these measurement errors consequential for DHS-based estimates of population-level characteristics? (Finaret et al. [2018]) suggests not in the case of stunting, but what about age?). Are they consequential for existing studies that use raw DHS data? The authors need to do more to convince the reader that this kind of measurement error matters.

Finally, I would agree with another referee that conflict needs to be explored as an important predictor of data quality (even if the paper does not move in the direction of a causal estimate on what drives measurement error). Even if left to a supplementary exercise (for example if conflict data do not match DHS survey years), conflict is an easily observable (and likely important) potential predictor of data quality for DHS operations. The authors state that AfroGrid data ends in 2020, but the Uppsala georeferenced conflict dataset extends to 2023: https://ucdp.uu.se/downloads/index.html#ged_global

More minor comments:

- 1) Page 5: It's unclear to the reader whether the Moran's I index was calculated on district-level estimates that are in turn aggregated from the 5x5 geostatistical model result. If that is correct, the high values for Moran's I are hardly surprising, as the structure of the geostatistical model presupposes spatial autocorrelation in order to create a surface interpolation. More appropriate, I think, would be a spatial autocorrelation test that uses the raw DHS data and cluster locations.
- 2) Line 124: "In some areas, pockets of comparable data quality spread across national borders. While this was partially a result of the supranational modeling framework we applied...": If the authors want to make a statement about whether data quality transcends national borders, this can easily be made into a formal test. The creation of the surface could have been done separately for each country, and a regression discontinuity framework can test for statistical differences in the data quality across international borders. An example of this method is Burke et al. (2016). Otherwise statements like this are too loose.
- 3) Line 130: "Furthermore, moderate to little correlation between the three data quality indicators themselves (Supplementary Fig. S3) resulted in distinct geographic patterns for each type of measurement error within countries.": This is somewhat puzzling for the reader. If these three variables selected from the authors are to be interpreted as broad proxies for data quality, what explains the low correlation between them? Or are these three measures picking up different kinds of data problems? In fact, that correlation coefficients are not so small (.49, .71, and .35).
- 4) Line 150: Please provide a citation for a DN of 15 being "above the threshold of the luminosity of public streetlights"
- 5) Line 190: Why is contraceptive use a "related public health indicator" to women's age?
- 6) Line 335: The paper refers a few times to "data quality predictions", I would make clear to the reader that these are the predictions generated in Result 1. In fact this sentence could go after the explanation of the geostatistical model so that the reader better understands what is being referred to when authors explain that result 2 is not using the output of that model.
- 7) Figure S2: Is the y-axis share of observations in DHS clusters? This is unclear to the reader.

Works Cited

Burke, Marshall et al., 2016, "Sources of variation in under-5 mortality across sub-Saharan Africa: a spatial analysis," *The Lancet Global Health*, Volume 4, Issue 12, e936 - e945.

(Remarks on code availability)

Reviewer #5

(Remarks to the Author)

I have read this manuscript with interest. The authors document interesting facts about the prevalence of three indicators of data quality in DHS data. DHS are cross-sectional data that cover many LMICs, often for several years, and are very widely used in different fields, including demography, public health, and economics.

- There is of course a vast literature on measurement error (ME) in survey data, also in LMICs. That ME is often made worse when respondents have low numeracy / literacy is also well known. One study in particular that the authors should cite is Larsen, Headey, and Masters (2019). "Misreporting Month of Birth: Diagnosis and Implications for Research on Nutrition and Early Childhood in Developing Countries", *Demography* 56:707–728. This paper, like this manuscript, analyses ME in many DHS data set, in particular (unlike this manuscript) in relation to errors in reported age in month of children. The paper proposes and tests concrete explanations for the patterns of ME, and shows how ME can have important implications for the estimation of causal effects of events that require exact information on the time of birth of children. Also because of this paper I think you need to tone down the statement that (267-68) "While survey experts or field enumerators may have an intuitive understanding that, for example, age-related information is hard to collect among remote populations, the exact magnitude of the problem has NEVER been assessed." Larsen et al indeed evaluate the exact magnitude of the problem, and go actually beyond that by finding explanations and exploring consequences. One key difference relative to your work is that they do NOT look at fine-grained variation in ME, although they do look at cross-country variation (also, they look at child age, while you look at adults). Measurement error in respondent's age in DHS is also analyzed in Rosenzweig (2021), "Age is measured with systematic measurement error in developing country surveys: A diagnosis and analysis of consequences" see <https://journals.sagepub.com/doi/10.1177/20531680211044068>. This also uses data from DHS.

- I am confused by the idea that "measures of sampling error" are "commonly used to indicate data quality in the literature" (bottom of p4). I do not think this is true. Sampling error usually exists even with no measurement error, and even if data are being collected with extreme care. Sampling error is not about "data quality" but rather about statistical precision, a very different concept. Of course you know this, and you write (263-264) that "[u]ncertainty of estimates and data quality itself are two different concepts and measuring statistical uncertainty alone, which is the common practice, does not necessarily address low data quality." This is of course true, but frankly I also think this is obvious and I don't recall ever seeing confusion about the two concepts, so I do not find this result very interesting and I see no reason why one should have strong priors about the relationship between sampling error and ME. Related: in your Result 3 you write that "standard deviations are commonly used as a measure for the statistical uncertainty of key public health measures," but standard deviations are just by definition a measure of statistical variability, so the statement above seems trivial (a similarly trivial statement would have been that "sample means are commonly used to measure the average of key public health measures"). Also, and related, you write that "sampling errors can often be statistically evaluated and corrected" (p8, line 198). While sampling errors are routinely evaluated, I don't think you can say that they can be "corrected". What you can do is express uncertainty in terms of sampling errors, e.g. through confidence intervals, but the sampling error remains and is not "corrected". Perhaps you can say that there are methods to "correctly" take them into account when gauging the degree of uncertainty around estimates.

- That your measures of data quality deteriorate with "remoteness" is perhaps interesting, but remoteness is surely correlated with many other factors, especially literacy / numeracy of respondents. You are careful not to make causal claims about the correlations you document, but on the other hand it was not clear to me why "remoteness" was chosen as a key correlate of ME.

Minor issues / typos

- P3, line 49, should be "Practitioners and researchers who work extensively with DHS data..." (drop "rely on"). Line below "data errors can be DISTRIBUTED ACROSS regions" sounds awkward / unclear, please rephrase.
- P4, line 62, should be "for example AT the level of districts"

(Remarks on code availability)

Version 2:

Reviewer comments:

Reviewer #4

(Remarks to the Author)

The authors have adequately addressed my comments.

Two small quibbles that I leave as optional for the authors:

- 1) Line 155: "5.4% above" should be "5.4 percentage points above". 20% to 25.4% is a 20% increase, in fact.
- 2) Line 380: "However, the predictors that we use likely capture many of these risks and are also likely correlated with

conflict." I find this sentence frustratingly imprecise. Given that rich geolocated conflict data exist, one could easily test whether the predictors you use actually predict conflict. Better yet, you could have added conflict as a predictor and reported in the paper or in the response to the referees whether or not model fit improved. Suggest, at the very least, that the sentence beginning with "However..." on line 380 be removed.

(Remarks on code availability)

Reviewer #5

(Remarks to the Author)

No further comments. Thanks for addressing my requests.

(Remarks on code availability)

Response to Reviewers | Household Survey Data Quality

In this document we compile all reviewer comments. Reviewer comments are summarized in bold and followed by direct quotations and summaries of the reviewers' concerns.

Reviewer 1

1. **Reference and use of relevant research.** Reviewer 1 had concerns that some key research was left out of the literature review and motivation section for this paper, particularly the development economics research on measurement error. Reviewer 1 stated, *"This is mentioned as it seems that the current paper seems unaware of this work and tackles the issue of measurement errors only from a health / demographic perspective (illness and malnutrition), and seems to suggest that policy makers are unaware of data gaps and data divides. As such, it misses much that has been learned in development economics."* Reviewer 1 provided a list of helpful references.

Thank you for your comment. We agree that the study would benefit from additional references to existing work on measurement error and data quality in the development and agricultural economics literature. We found particularly interesting studies that estimate measurement error in microdata and agricultural surveys. Among others, key references in the background/literature review section include works by Devarajan (2013), Jerven (2013), Naude (2014), Adebisi and Lucero-Prisno (2022), and Gibson & Kim (2010). Due to limited space in the main manuscript, we provided a brief discussion of this literature in the Supplementary Information. These additional references exemplify existing explorations of data quality in fields beyond health and demography, which is our focus. The significant additions to the literature review/background have enhanced the manuscript and made its specific contribution clearer.

2. **Incorporating conflict as a variable in the model – conflict as a blind spot in the model.** Reviewer 1 stated, *"Certainly, many authors working on conflict in SSA, the nature of vulnerability and fragility on the continent, and the impact of conflict on health, poverty, and migration, would strongly argue that the authors miss a very important spatial determinant of outcomes, but moreover a variable that affects the quality of data collected - in obvious ways (and perhaps this is even more important in Central Africa, where the authors found the best predictive performance of their model....)." Hence, it is not entirely accurate as the authors claim that "we still lack knowledge about the spatial distribution and magnitudes of data errors where it matters most - on subnational and district levels." The studies cited have resulted in good knowledge about measurement errors in agriculture, the role of conflict in measurement errors, etc."*

Thank you for this comment. We agree that the presence of conflict and civil insecurity is likely to affect data quality. We did not see it as feasible to include because the AfroGrid data ends in 2020, and the vast majority of surveys we include are now from 2021 and 2022 after updating the DHS surveys. We are also concerned that the conflict data are relatively volatile and are subject to measurement error themselves (i.e., missing events, also difficult to connect the timing and spacing of conflict with the timing and spacing of the surveys accurately). In addition, a consensus-driven and generalizable "conflict exposure" variable does not yet exist, although we do think that there are some interesting research questions about this issue and how it intersects with data quality. We expect that such a data linkage with conflict data might be possible for a particular region if the conflict data were particularly rich and detailed in terms of timing, and if we felt we could develop a spatially aggregated indicator of conflict exposure. However, this is

beyond the scope of our current study. We have removed the phrase “we still lack knowledge,” which as reviewer 1 has pointed out, is not entirely accurate. We have enhanced the discussion of the development/agriculture economics literature as described above. We also know that DHS does not operate in regions or countries where it is unsafe for survey enumerators, such as in Somalia.

- 3. Methodological issues of no longitudinal data and correlation between conflict and measurement error in the variables of interest:** Reviewer 1 stated: *“The only methodological issues I would flag is the likely high correlation between conflict across space and measurement errors in their variables of interest as already mentioned. Related to this, is their lack of using longitudinal data, which has typically been a central strategy used by economists to reduce the impact of unobserved heterogeneity across time where these have been time invariant. For instance, some of the geo-variables that the current study control for, such as terrain ruggedness, are time invariant; to the extent that it would affect nutritional outcomes, using longitudinal data would control for this. The changes in measurement errors over time is therefore a non-trivial issue and somehow not discussed.”*

Thank you for this comment. We think that it would be methodologically useful to examine changes over time in data quality, and to use fixed-effects for time-invariant factors which may influence data quality like terrain. This would allow us to explore more about the time-variant factors and their role as determinants of data quality, particularly conflict and civil insecurity in some areas. As currently stands, our model is examining the most recent survey in each included country and thus is a cross-sectional model, which we find useful to give a snapshot in time of an important construct (of data quality). In the future, we plan to extend our analysis to include longitudinal data and answer questions about changes in measurement error over time. A full panel data model may be computationally infeasible due to the significant increase in the amount of data used, but we agree that it would be a relevant future research pathway. Additionally, a cross-sectional approach has been used because many countries only had data available for one round of the Standard DHS. A longitudinal analysis would exclude those countries. We do show that data quality is relatively stable across time, for those countries with multiple surveys available in result 2 (Figs 9-12 in the Supplementary Information). Therefore, while this repeat cross-sectional analysis is not as informative as a panel study, it is unlikely that the results are due to cross-sectional variation at a point in time.

- 4. Addressing the “so what” question:** Reviewer 1 stated, *“Finally, given that it is already well known that data divides and gaps exists, that remote populations are more vulnerable as a result, and that work on measurements errors in surveys are growing and ongoing, the questions that the current paper do not fully answer are "so what? what should researchers and policy makers be aware of? Can their methods shed light on how to reduce measurement errors?" The policy and research implications should be highlighted, including being stated upfront. Otherwise the reader will come away from the paper impressed by the data analytics, but with the feeling that not much has been found that was not already known or can be reasonably assumed.”*

Thank you for making this point. We have revised the text to highlight the relevance of the findings and the implications for policy and research in the beginning of the paper, and clarified the implications of the study in the discussion section of the paper. Here we will clarify the implications of our paper and we have included this explanation in the manuscript as well.

The main motivation for this study is that even practitioners and researchers who work a lot with DHS data don't know how large the variation in data quality really is. The most recent DHS methodological report on this issue described implausible values and concluded that "future research is needed to examine variation in team performance at the regional level to explore the impact of data quality on the interpretation of anthropometric estimates" (Benedict et al. 2022, DHS Methodological Report 33, p. 44 and 45). We take this as indication that while DHS – like many other researchers and data users as well – are aware of a potential problem, our work is the first to estimate the extent of the errors at district level and at finer resolutions. The larger part of researchers, journals and reviewers simply trust DHS and sometimes look at standard deviations (for measures of uncertainty). We aim to raise awareness with this latter group and quantify the problem for the former group.

One of the main implications is that (1) future DHS rounds can improve in areas where we report low data quality – a recommendation raised in many DHS reports including the most recent one (MR 33). This potential for improvement in key geographic areas (and focused on specific challenges – like reducing non-response if that is what is needed in the region) is an implication that we discuss in the manuscript but have revised for clarity in the discussion section. Another implication is that (2) practitioners can understand more about the quality of the DHS data that they are relying on, in the moment, because they often rely on DHS surveys for micro-level data about their populations. With better DHS data also in remote areas, more practitioners would also be able to use them. Another implication is that (3) researchers, journals, and donor agencies will benefit from more awareness and caution when inferring causalities based on DHS data, especially in remote areas. Finally, for applied researchers in particular, another implication is that (4) it is very clear that data quality needs to be considered when analyzing data from remote populations. These implications come directly from our results, which show that relying on just reporting statistical uncertainty is inadequate because there is a bias in the distribution (result 2) and because standard deviations and our reported data errors do not overlap (result 3). These implications are now all made clearer throughout the discussion section as requested by Reviewer 1.

Reviewer 2

- 1. Clarity of Presentation:** Reviewer 2 stated, *“The manuscript could benefit from improved clarity in its presentation of analysis results on the distribution of errors. Some paragraphs are densely packed with information, making it challenging for readers to follow the arguments and findings. Clearer organization (such as using tables) and more concise language would enhance readability.”*

Thank you for this comment. In revision we have been able to clear up certain spots where the language was not as concise as possible. We have also kept various results to support the main findings in the Supplemental Information so that the manuscript itself is not too long.

- 2. Methodological transparency:** Reviewer 2 stated: *“While the manuscript discusses various quality indicators, there is room for greater transparency in the methodology used for data analysis and the justification in selecting the type of survey (DHS over MICS), geography (Sub-Saharan over all countries worldwide), year of survey, and subnational level of analysis (level-2 admin over level-3 and level-4 admin for some countries). Providing more details on the criteria used for assessing data quality from these perspectives would enhance the manuscript's rigor.”*

Thank you for this comment. We have chosen to focus on Sub-Saharan Africa because DHS data often serve to fill data gaps in many countries in Sub-Saharan Africa, and the DHS program is the most widely used household survey program in the world. As stated in the Methods section: “DHS data are best suited for the scope of the study, because they are publicly available, geocoded, and more comprehensive in time and space than data from other large survey programs such as MICS or LSMS.” In addition, several of the public health outcomes of interest are of particular concern in Sub-Saharan Africa, such as child stunting and contraception use, making it an important region to study the quality of data used to build these indicators. We provide detailed background information on the data quality indicators and how they were developed. In the interest of space in the manuscript, some of these details are located in the Supplemental Information.

- 3. Using updated surveys:** Reviewer 2 stated, *“Given DHS surveys have become more robust in terms of sample size and methodology over time, it is not clear why authors chose to limit their analysis to surveys between 2006-2016 (for some countries goes to 2018/19). For the results to be more informative, as the study suggested, countries with recent DHS survey data with geographic details and perhaps with more sample size and greater reach to remote areas need to be incorporated in the analysis.”* Reviewer 2 provided a list of the countries with more recent surveys.

Thank you for this comment. We agree that using the most up-to-date surveys available is ideal for this project, and we have now included them. The data now cover the most recent Standard DHS surveys through 2022. We have also been able to expand the number of included countries to 35 with Gambia and Mauritania as new additions. Some of the surveys that Reviewer 2 has suggested were not the Standard DHS surveys, such as for maternal health or were “mini” or interim surveys, and we did not include those. These “mini” surveys are smaller in scope, without sufficient data for our model which is the reason why we could not include them. However, all standard, country-wide DHS survey rounds are included.

- 4. Addressing limitations:** Reviewer 2 stated: *“The manuscript should acknowledge its own limitations and potential sources of bias. An honest appraisal of the study's limitations would enhance the manuscript's credibility and assist readers in interpreting the findings.”*

Thank you for this comment. Our limitations section has been updated to describe additional study limitations, and now reads as follows: “Our modeling framework is, like any statistical model, an abstraction of reality and subject to several limiting simplifications. Spatial data gaps between DHS survey locations may introduce varying degrees of uncertainty in our estimates. These considerations also apply – to a lesser extent – to the aggregated analyses and are relevant to any geolocated household-survey based data set. Also, survey locations are randomly displaced in space for data confidentiality reasons, further increasing the level of spatial uncertainty. However, the covariate layers we use to extract external information for each DHS cluster do not vary substantially at this high resolution and we expect that our overall findings will remain essentially unchanged. Other limitations include that we choose one model specification out of many, which holds potential influence on the predictions. Our estimates include uncertainty which we do not show, as the maps are merely point estimates.”

5. **Practical applications:** Reviewer 2 stated: *“While the manuscript offers recommendations for improving survey data quality in future, it would be valuable to provide concrete examples of how these recommendations can be applied in real research scenarios. This would assist readers in translating theory into practice. For instance, most of these countries have started collecting routine facility data, and therefore, it would be more helpful to expand the discussion on how to leverage such datasets as appropriate, particularly for policy, planning and programming purposes. Furthermore, it is likely that many development-related factors (such as increased access to roads, education, and telecommunication, etc.) brought opportunities to increase the geographic reach of DHS sampling. Given most countries have multiple surveys over the past two decades, it would be worthwhile to expand the discussion using results from Figure S9 to highlight whether or DHS sampling strategies took advantage of increased access to remote areas.”*

Thank you for this comment. We agree that concrete applications are important. We think that the online visualization tool we built (https://apps.worldpop.org/SSA/data_quality/) will be especially useful for enumerators and data managers in the field and in the office to explore previous data quality challenges in the area that they are working. To address this reviewer concern, which overlaps with a concern from Reviewer 1, we also enriched the discussion of the practical implications of the findings in the discussion section of the paper.

Reviewer 3

1. **DHS surveys were not designed to explore errors or data at such a fine geographic specificity:** Reviewer 3 states, *“Extrapolating those errors down to a level of geographic specificity that the DHS does not (and is arguably not designed to) provide. My doubts here are less with the method used to impute errors at, say, the district level, and more with the motivation and utility of doing so.... Yes, policy decisions happen on a sub-national basis, of course. But DHS data users are typically discouraged from producing or using estimates at the district level. The DHS program, as the authors know very well, typically only claims the surveys are representative at a higher administrative level, typically admin-1 or even higher. It would be useful for the paper to illustrate the kinds of decisions currently being made on the basis of DHS data at the district level. At the risk of overstating my point, at present this section reads to me as ‘here’s something you were never supposed to do with DHS data, and sure enough, it seems dubious to do so.’”*

Thank you for this comment. We have clarified this important point in the text, describing some of the spatial modeling that is used for various health indicators from the DHS and elsewhere. While the DHS program was not designed specifically to develop modelled surfaces, the DHS Program themselves routinely produces modelled surfaces using similar methods at the grid-square level, as can be seen, for example, on this website: <https://spatialdata.dhsprogram.com/modeled-surfaces/>.

There is also a growing amount of work using these spatial approaches including the Malaria Atlas Project which is used by the World Health Organization, and we have now included examples like this into the revised manuscript to highlight this increasing usage. We would argue that using cluster-level DHS data in empirical analysis is an increasingly common practice. The DHS Program itself also produce contextual variables at the DHS cluster level – DHS Spatial Covariate data files for download, so that researchers can merge these spatial covariates into the DHS survey data. Moreover, the MICS program at UNICEF are

continuing to develop similar outputs, including geospatial measures at the cluster level (e.g. <https://www.youtube.com/watch?v=WITZG-KvgjY>).

In the realm of nutrition specifically, Buckland et al. 2020 argue through an extensive survey that “Periodic, population-based surveys form the backbone of most national nutrition information systems. Commonly available household surveys include the DHS, MICS, as well as a variety of national and sub-national surveys...” Buckland et al. (2020) also find 82% of their respondents did not have access to data at the level of geographic disaggregation that would be necessary to support their work. In addition, the 51st Session of the Committee on World Food Security in 2023 stated that “governments should strive to.... Increase and sustain responsible investment and adequate resources for the production of timely, quality, disaggregated, where relevant, reliable and consistent (food security and nutrition) data...”

Another benefit of examining data quality at this resolution is that many researchers use DHS cluster level data to estimate the impact of climate or conflict shocks on cluster-level outcomes, so they would benefit from knowing about a finer resolution of data quality. Therefore, even if we cannot assume that cluster-level indicators are representative, researchers and the DHS Program itself uses cluster-level data in many other kinds of ways for which this work on data quality becomes relevant. In general, given that many countries are working to track progress towards the Sustainable Development Goals, using modeled surfaces like in this paper is an essential strategy to fill data gaps and explore how data quality can be improved in the future.

2. **Clearer analytical statement of the result:** Reviewer 3 states about Result 1, *“Finally, while the maps are useful, it would also be nice to have a clearer analytical statement of the “result” here. For instance, is the variation higher than would be produced by chance? I struggle to translate the finding into concrete words.”*

Thank you for this comment. We agree that there should be a clearer analytical statement of the result here. We have therefore estimated the Moran’s I statistic for all three indicators at the district and grid levels. The Moran’s I statistic measures spatial autocorrelation and its values range from -1.0 to +1.0. The null hypothesis when using Moran’s I for inference is that the geographic attribute being measured is randomly distributed. The results are now in the text as follows: *“We see substantial spatial autocorrelation across neighboring districts, summarized using Moran’s I at the district level for age heaping ($I=0.971$, $p<0.0001$), the incomplete age information for women (0.896 , $p<0.0001$), and flagged HAZ values ($I=0.949$, $p<0.0001$).”* Therefore, the use of Moran’s I in the case of this study confirms the non-randomness of data errors across space. Tables S10-S12 provide all Moran’s I statistics at the district level with p-values indicating the variation in spatial autocorrelation.

3. **The term ‘measurement bias’ is unclear:** Reviewer 3 states, *“The section refers to ‘measurement bias’, but I’m unclear exactly what bias this is. If analysts use the DHS to compute regional estimates of, say, child mortality, would the patterns of measurement error documented here bias the results somehow? Perhaps so, but it doesn’t seem obvious to me what kind of bias there would be. The paper would benefit from some analytical account of this, and/or an illustration. (As an aside, in line with my comments on result 1, I can’t resist noting that the authors rely on raw DHS data, rather than their model imputation above, to study the relationship between remoteness and errors. This reinforces my doubts about the need for such imputations of error levels at high levels of geographic disaggregation!)”*

Thank you for this comment. We agree that the term “measurement bias” was misleading. We have changed wording and clarified that what we are studying is a bias in the geographic distribution of errors in DHS raw data. We have also revised it to “data quality” and have defined data quality in the introduction as: *“We define data quality as a combination of high precision and high accuracy, meaning that measurements have low random error and low systematic error.”* We have clarified in the methods section why we prefer using DHS raw data in result 2: *“We used raw DHS data to avoid reporting a purely artificial, mechanic relationship arising from regressing data quality predictions (that are a function of distance to nightlight) on distance to nightlight.”*

- 4. Clear statistical statement of the finding (e.g., a correlation):** Reviewer 3 states, *“I was surprised (in a good way) by the suggestion that measurement error and sampling uncertainty are not concentrated in the same geographic places. As with result 1, however, I didn’t quite see a clear statistical statement of this finding. Even a simple correlation coefficient would help (if this is reported and I missed it, apologies). Once again, the maps are an imperfect substitute for a clear, unambiguous statement or statistic that can be tested or compared to findings from elsewhere.”*

Thank you for this comment. We agree that a clear statistical result of the finding for Result 3 would improve clarity and provide helpful additional information to the maps. Therefore we have estimated correlation coefficients for each indicator of data quality (missing data and standard deviations) for each included indicator (contraception use and stunting prevalence). The results were very similar at the grid and at the district level. In the main article we report on district level results which now reads in the text as follows: *“We estimated correlation coefficients between the two types of data quality for each indicator at the district level. Missing data for women’s age was moderately negatively correlated with the standard deviation of contraception use ($\rho=-0.402$, $p<0.0001$). Flagged HAZ values were slightly positively correlated with the standard deviation of the prevalence of stunting ($\rho=-0.109$, $p<0.0001$).”* Figure S15 in the Supplementary Information provides the related charts. We concluded that the missing data and the standard deviation of data quality indicators are not strongly correlated at the district levels.

- 5. Clarity of meaning for Result 3:** Reviewer 3 states, *“Lastly, while interesting for someone like me who often uses DHS data, the motivation for exploring this correlation was again not super clear to me. I am tempted to treat the (implied) low correlation between errors and uncertainty as good news, spreading the source of confusion more evenly across geographic space. But that’s a casual speculation, and I’d benefit from more concrete explanation of what we should conclude from this.”*

Thank you for this comment. Our reason for showing Result 3 is that commonly used standard deviations to explore precision do not necessarily address the whole story when it comes to data quality. Low precision/uncertainty as indicated by a high standard deviation of an indicator is a different problem (stemming for example from survey design and sample size) than low data quality (data errors such as measurement errors or missing data). We concluded in the discussion that *“uncertainty of estimates and data quality itself are two different concepts and measuring statistical uncertainty alone, which is the common practice, does not necessarily address low data quality.”*

Of those two threats to analytical work, analyzing data quality particularly challenges researchers who are using secondary data and for which they cannot go collect detailed information on sub-samples to assess systematic bias, for example. Consulting the map for which type of data quality problem (uncertainty,

error, or both, or neither) would help them assess what they needed to do in their own modeling and research.

REVIEWER COMMENTS (received Sept 25th 2024)

We would like to thank all the authors who took the time and effort to suggest extremely useful corrections. We have also revised substantial parts of the manuscript and believe it has improved significantly. Thank you for your contributions.

Reviewer #1 (Remarks to the Author):

In my opinion they have adequately addressed these, and I am satisfied that the paper is suitable for publication.

Response: Thank you. The manuscript has been improved thanks to your comments and suggestions.

Reviewer #4 (Remarks to the Author):

Dear editor,

I am writing to peer review article number NCOMMS-23-43116A, titled "Subnational variations in the quality of household survey data in sub-Saharan Africa".

This paper quantifies spatial variation in data quality in the most recent Demographic and Health Survey of 35 African countries, focusing on incomplete age, age heaping, and flagged HAZ as three examples of poor data quality. The paper then creates 5 km x 5 km surfaces of these indicators covering the 35 countries in order to help users of DHS data in academic and policy spaces know which regions of which countries likely have poorer data quality. Finally, the authors show that data quality worsens at longer distances from nightlights (using a DN 15 as the threshold), and they show that there is no correlation between their measures of data quality and measures of statistical uncertainty from sampling. I will first offer more overarching comments and then offer more minor comments.

First, I enjoyed the paper and the focus on spatial variation in data quality in the DHS. I concur with the authors that the DHS has emerged as a very frequently used data source not only in academia but also in development and policy for within-country targeting of resources for health and development. Any novel work on the DHS that affects its use by this wide audience is therefore of important general interest. I therefore welcome the idea of quantifying measurement errors at subnational level for a large number of countries for users of the DHS.

My first question regards what follows the primary contribution. The paper could have gone in the direction of exploring what drives differences in data quality, as another referee has suggested, but this causal framework would call for different kinds of statistical analysis (exploiting longitudinal data, fixed effects, etc.). The authors have chosen not to go in that direction, which I don't think is a problem, but it does mean that the remaining contributions of the paper should be substantive. The authors offer two add-ons to the surface mapping of measurement error: **(a)** they document that the

problem becomes more severe as distance from urban areas (proxied by NTL) increases, and **(b)** the sampling variation is uncorrelated to the measurement error.

Regarding (a), as the authors already state, it is already known that "...[DHS] data in remote areas can be less reliable relative to urban areas" (line 51). If the result is not too novel, perhaps the authors can deepen the analysis by adding other variables predictive of measurement error. These could be household assets and characteristics, ethnicity and language, etc. Knowing whether other variables observable to DHS enumerators and DHS users are predictive of measurement error would be useful for targeting additional resources to improve data quality, or for analysts, to statistically account for lower data quality using other DHS variables.

Response: Thank you for your comments. Indeed, we are not aiming for a causal analysis and we agree that this means the remaining contributions must be substantive. We believe our statement that "*...it is already known that [DHS] data in remote areas can be less reliable*" was formulated too strongly, as no spatial quantification of data problems has ever been conducted at the subnational level. We have rewritten this section of the introduction to better place our work (a spatial quantification of data errors) in the existing literature (only country / survey level), and to better highlight the relevance for those researchers and practitioners who work a lot with DHS data, and those who so far have taken DHS data without much knowledge of potential limitations. For experienced researchers, they can get the first ever subnational estimates of DHS data errors from this paper. For researchers who are less experienced with DHS data, they can benefit from the maps in Result 1 which illustrate the extent of the problem for three most used variables.

(a) After much deliberation in the team, we decided not to add additional variables. Instead, we worked on softening our statements, e.g., "it is already known that...", which was too strongly worded. We now clearly state what is already known (e.g., that there are data errors in more remote areas) and what is not yet known (the extent of these errors and where they occur using 5x5 km grid estimates). We have made this distinction clearer in the manuscript by stating, "*As a result, even those practitioners and researchers who work extensively with national household surveys and who are aware of the potential for spatial variation in data quality, do not know the precise spatial distribution and the magnitude*"

We use distance to NTL DN15 and distance to urban settlements, because they capture the effect of many known potential factors such as household assets, institutional quality etc and our intent is to illustrate the vast spatial variation in data quality and that data quality deteriorates non-randomly. We hope that future work in this area will be able to disaggregate the effect of specific variables predictive of measurement errors which would be a significant extension of the presented statistical analysis.

Our main point is that there is a measurable spatial bias in data errors. Testing a set of variables that explain this at the subnational level, in our view, requires a significantly extended analysis, because these variables are highly correlated and the relationships are complex. We therefore use NTL and distance to settlements as they capture a number of potentially predictive variables.

Regarding (b), the authors could improve the motivation. On line 201 they explain that result 3 was explored because they were "curious to know to which extent high sampling uncertainty of stunting overlapped with the biologically implausible height-to-age measurements taken in the same population." A better motivation would explain why answering this question is important, and how the overlap (or lack thereof) is important for users of the DHS. High statistical uncertainty in rural places with small population (and sample) sizes is to be expected and is in line with the DHS sampling strategy. So concluding that more sampling is needed there is counter to the DHS strategy. Moreover,

investing in increased sample size and improved data quality might be at odds from a resources point of view. In summary, the documented correlations between uncertainty and measurement error, therefore, need to be better motivated for the reader.

- (b) Thank you for this comment. We agree that the motivation for developing result 3 should be improved with some clarification. Therefore, we have replaced the phrase "*...curious to know...*" with a more clearly articulated argument placed at the end of the introduction, for why this study is important to DHS users and the DHS program. Our motivation for result 3 is not simply to suggest increasing the sample size in areas of higher statistical uncertainty.

In the introduction, we define data quality quite narrowly - as having both low statistical uncertainty and as little collection bias in measurement and missing data as possible since each problem can affect the insights that can be drawn from household survey data, even if in different ways. Therefore, result 3 suggests that future assessments of data quality may benefit from looking at both dimensions of data quality, not just uncertainty, which may well be the result of DHS's sampling strategy.

A second question regards the authors' assertion of bias generated by the measurement error. For example on line 147 when discussing the second result, the authors argue that lower data quality in more rural areas generates a bias. Similarly, the paragraph beginning on line 265 discusses "non-randomly" distributed errors documented by the study. While this could be true if calculating a national estimate of population statistics, the relevant use case here is for localized statistics. For this purpose, it's unclear to me why there is necessarily bias, since missingness occurring randomly within the data distribution of rural observations (even if more frequent than more urban observations) would not generate a bias, only noisier estimates. Just because there is more missing data in rural areas does not mean it is non-random within the rural sample, which is what would affect bias in calculating population statistics for that region, or causal studies on household-level data.

Response: Thanks for pointing this out. We have clarified in result 2 and in the discussion why we believe our findings are relevant also for smaller than national geospatial estimations. First, we also consider noisy, uncertain estimates as problematic for drawing insights (see our definition of data quality) and second scholars studying remote populations will be made aware of different potentials for the distribution of data errors across countries and across DHS variables (Figure 2). We have made this a stronger argument in result 2 and in the discussion. Thanks for this important point.

In fact, the authors cite a study by Finaret et al. [2018] which finds that bias from unusable HAZ data is likely small. This then begs the question: how important is this issue that the authors are documenting? Are these measurement errors consequential for DHS-based estimates of population-level characteristics? (Finaret et al. [2018]) suggests not in the case of stunting, but what about age?). Are they consequential for existing studies that use raw DHS data? The authors need to do more to convince the reader that this kind of measurement error matters.

Response: Thank you for this comment. We believe that result 2 is important for data users, as it is the first subnational analysis of remoteness affecting three different data quality indicators. We find that the "remoteness penalty" impacts DHS variables to different extents. It is a novel finding that missing data and measurement errors change in different patterns with increasing distance to settlements. HAZ (stunting) is the least affected variable, as we expected based on the findings of Finaret 2018. Compared to Finaret et al. 2018, who conduct cross-sectional analysis comparing surveys – assessing one variable – HAZ – for its quality, we extend and deepen the analysis of data errors at the subnational level and find that two other indicators deteriorate more intensely with increased distance to settlements. Our findings may motivate

data users who study remote populations to assess the quality of DHS data the work with on a “case by case” or “variable by variable” basis as we find different extents across our three data quality indicators (which are built on DHS data variables). We clarified this in the introduction of result 2 and towards the end of the discussion.

Finally, I would agree with another referee that conflict needs to be explored as an important predictor of data quality (even if the paper does not move in the direction of a causal estimate on what drives measurement error). Even if left to a supplementary exercise (for example if conflict data do not match DHS survey years), conflict is an easily observable (and likely important) potential predictor of data quality for DHS operations. The authors state that AfroGrid data ends in 2020, but the Uppsala georeferenced conflict dataset extends to 2023.

Response: Thank you for this suggestion. In response to a previous reviewer, we explored conflict as a predictor and figured that it does not add much additional variation over the predictors already included in the model. Of course, we agree that conflict limits the reliability of household survey coverage, not least because of the inability to sample in unsafe and insecure areas. As a result, DHS enumerators omit conflict-affected areas in the national rounds of the survey that form the basis of our analysis. In addition, the predictors we use are likely to capture many of these risks and are also likely to be correlated with conflict. We acknowledged this in the manuscript and included studies of supporting evidence: “*Conflicts limit the reliability of coverage from household-based surveys due to an inability to sample in unsafe and insecure areas.*”

New evidence/references:

Sbarra, A. N. *et al.* Estimating vaccine coverage in conflict settings using geospatial methods: a case study in Borno state, Nigeria. *Sci Rep* **13**, (2023).

Karaye, I. M., Stone, K. W. & Horney, J. A. Determinants of Under-Five Mortality in an Armed Conflict Setting: Empirical Findings from the Demographic and Health Surveys. *Int J Environ Res Public Health* **19**, (2022).

Schon, J. & Koren, O. Introducing AfroGrid, a unified framework for environmental conflict research in Africa. *Sci Data* **9**, (2022).

Wagner, Z. *et al.* Women and children living in areas of armed conflict in Africa: a geospatial analysis of mortality and orphanhood. *The Lancet Global Health* **7**, (2019).

More minor comments:

1) Page 5: It’s unclear to the reader whether the Moran’s I index was calculated on district-level estimates that are in turn aggregated from the 5x5 geostatistical model result. If that is correct, the high values for Moran’s I are hardly surprising, as the structure of the geostatistical model presupposes spatial autocorrelation in order to create a surface interpolation. More appropriate, I think, would be a spatial autocorrelation test that uses the raw DHS data and cluster locations.

Response: Thank you for this valuable comment. We recalculated Moran's I with raw data and found still considerable but varying values across indicators and across data. We updated the text and the corresponding tables in the Supplementary Information.

2) Line 124: “In some areas, pockets of comparable data quality spread across national borders. While this was partially a result of the supranational modeling framework we applied...”: If the authors want to make a statement about whether data quality transcends national borders, this can easily be made

into a formal test. The creation of the surface could have been done separately for each country, and a regression discontinuity framework can test for statistical differences in the data quality across international borders. An example of this method is Burke et al. (2016). Otherwise statements like this are too loose.

Response: Thank you for this comment. We removed this section starting with “*In some areas...*” to reduce confusion about the motivation of the present paper and remove the loose language. We believe that examining trends and breaks at national borders and across DHS teams would be an interesting follow-up project, especially now that we have information on the subnational distribution of measurement errors. We also added the Burke et al. (2016) paper as a reference in this paragraph: “*Following recent studies that have revealed considerable heterogeneity in important health and nutrition indicators of human well-being at the subnational level, which had previously been masked by country-wide measures, we hope to motivate more research into the causes of the subnational heterogeneity in data quality, including low birth registration, low access to education, migration, or high levels of poverty.*”

New reference:

Burke, M., Heft-Neal, S. & Bendavid, E. Sources of variation in under-5 mortality across sub-Saharan Africa: a spatial analysis. *The Lancet Global Health* **4**, (2016).

3) Line 130: “Furthermore, moderate to little correlation between the three data quality indicators themselves (Supplementary Fig. S3) resulted in distinct geographic patterns for each type of measurement error within countries.”: This is somewhat puzzling for the reader. If these three variables selected from the authors are to be interpreted as broad proxies for data quality, what explains the low correlation between them? Or are these three measures picking up different kinds of data problems? In fact, that correlation coefficients are not so small (.49, .71, and .35).

Response: Thank you for this comment. We corrected our wording and expanded this section to better explain that the underlying errors and patterns in the errors differ because data challenges are not the same across these indicators. We added that “*There was little to moderately positive correlation between the three data quality indicators. The correlation coefficients between pairs of indicators ranged from 0.35 between age heaping and incomplete age to 0.71 between the two age-related indicators, age heaping and flagged HAZ (see Supplementary Figure S3). Data quality challenges are not uniform across the indicators we employed, and different correlation coefficients reflected different underlying potentials for systematic measurement error and missing data. ... The only moderate and little correlation between the three data quality indicators resulted in distinctive subnational spatial patterns for each indicator. This reflects the varying degrees of difficulty encountered in collecting the different types of measurements across localities and across DHS data collection teams.*”

4) Line 150: Please provide a citation for a DN of 15 being “above the threshold of the luminosity of public streetlights”

Response: Thank you for this comment. We have added the relevant citations to the main manuscript. Before that they were only correctly cited in the Methods section: “*DN 15 is below DN 20 assumed for urban, built environments such as in smaller cities.*”

5) Line 190: Why is contraceptive use a “related public health indicator” to women’s age?

Response: Thank you for this comment. We added the following to better explain this relationship: “*...contraceptive use is an important health indicator for women, as it relates to*

their overall health and ability to achieve their desired fertility. In addition, maternal health status is a major determinant of stunting.“

6) Line 335: The paper refers a few times to “data quality predictions”, I would make clear to the reader that these are the predictions generated in Result 1. In fact this sentence could go after the explanation of the geostatistical model so that the reader better understands what is being referred to when authors explain that result 2 is not using the output of that model.

Response: Thank you for this comment. We agree that the paper would benefit from improving the precision of our writing and definitions. We have worked on this throughout the manuscript and changed between, e.g., “data quality predictions” to “predictions of data quality as generated in Result 1” or “errors found in raw DHS data,” etc.

7) Figure S2: Is the y-axis share of observations in DHS clusters? This is unclear to the reader.

Response: Thank you for this comment. The y-axis shows the minimum, maximum and the average (population weighted) values of data errors across DHS clusters. This was unclear before. We corrected the caption. Thanks for pointing that out.

Reviewer #5 (Remarks to the Author):

I have read this manuscript with interest. The authors document interesting facts about the prevalence of three indicators of data quality in DHS data. DHS are cross-sectional data that cover many LMICs, often for several years, and are very widely used in different fields, including demography, public health, and economics.

- There is of course a vast literature on measurement error (ME) in survey data, also in LMICs. That ME is often made worse when respondents have low numeracy / literacy is also well known. One study in particular that the authors should cite is Larsen, Headey, and Masters (2019). "Misreporting Month of Birth: Diagnosis and Implications for Research on Nutrition and Early Childhood in Developing Countries", *Demography* 56:707–728. This paper, like this manuscript, analyses ME in many DHS data set, in particular (unlike this manuscript) in relation to errors in reported age in month of children. The paper proposes and tests concrete explanations for the patterns of ME, and shows how ME can have important implications for the estimation of causal effects of events that require exact information on the time of birth of children. Also because of this paper I think you need to tone down the statement that (267-68) "While survey experts or field enumerators may have an intuitive understanding that, for example, age-related information is hard to collect among remote populations, the exact magnitude of the problem has NEVER been assessed." Larsen et al indeed evaluate the exact magnitude of the problem, and go actually beyond that by finding explanations and exploring consequences.

One key difference relative to your work is that they do NOT look at fine-grained variation in ME, although they do look at cross-country variation (also, they look at child age, while you look at adults). Measurement error in respondent's age in DHS is also analyzed in Rosenzweig (2021), "Age is measured with systematic measurement error in developing country surveys: A diagnosis and analysis of consequences" see <https://journals.sagepub.com/doi/10.1177/20531680211044068>. This also uses data from DHS.

Response: Thank you for this comment. This was a similar comment to the first comment from Reviewer 4 as well. Thanks for the additional reference as well.

We agree that important work has been done including assessing the magnitude of specific types of measurement errors and studying potential explanations and consequences. We have clarified this in the discussion and deleted the sentence in question (“*While survey experts...*”). We also made clearer that our contribution is the spatial quantification at the subnational levels, which has been lacking so far, not necessarily finding explanations. We have also added low numeracy / literacy as potential factor for data errors in the text, adding Larsen et al. 2019 as citation. We also used Larsen et al. 2019 to improve the description of “flagged HAZ” in the Supplementary Information. Finally, we explicitly mentioned their important findings again in the Discussion of the main manuscript.

We also added Cummins (2013) as another example of a paper which examines data errors in the DHS, specifically related to child age.

A key new paragraph addressing this reviewer comment is in the Discussion section: „*Survey experts or field enumerators who have studied measurement errors in DHS data have created awareness for the potential of systematic measurement errors and missing data using cross-country and cross-survey analysis to assess the variation in measurement errors, their magnitude, as well as potential factors and consequences. However, the spatial dimension of systematic measurement errors at fine-grained local levels, where decisions and planning is taking place, has been unknown.*“

In addition, throughout the study, we have made clearer that, different from Larsen et al. 2019 and similar studies, our contribution is the quantification of measurement errors at the subnational levels which has not been done before. We believe that cross-country and cross-survey measures of systematic measurement errors mask a heterogeneous spatial distribution as we aim to show in our results.

Thank you also for the Rosenzweig reference (2021). We have added this to our introduction and in the description of our indicator “age heaping” in the Supplementary Information.

New references:

Larsen, A. F., Headey, D. & Masters, W. A. Misreporting Month of Birth: Diagnosis and Implications for Research on Nutrition and Early Childhood in Developing Countries. *Demography* **56**, 707–728 (2019).

Rosenzweig, S. C. Age is measured with systematic measurement error in developing country surveys: A diagnosis and analysis of consequences. *Research & Politics* **8**, (2021).

Cummins, J. *On the Use and Misuse of Child Height-for-Age Z-Score in the Demographic and Health Surveys*, (2013).

- I am confused by the idea that "measures of sampling error" are "commonly used to indicate data quality in the literature" (bottom of p4). I do not think this is true. Sampling error usually exists even with no measurement error, and even if data are being collected with extreme care. Sampling error is not about "data quality" but rather about statistical precision, a very different concept. Of course you know this, and you write (263-264) that "[u]ncertainty of estimates and data quality itself are two different concepts and measuring statistical uncertainty alone, which is the common practice, does not necessarily address low data quality." This is of course true, but frankly I also think this is obvious and I don't recall ever seeing confusion about the two concepts, so I do not find this result very

interesting and I see no reason why one should have strong priors about the relationship between sampling error and ME. Related: in your Result 3 you write that "standard deviations are commonly used as a measure for the statistical uncertainty of key public health measures," but standard deviations are just by definition a measure of statistical variability, so the statement above seems trivial (a similarly trivial statement would have been that "sample means are commonly used to measure the average of key public health measures"). Also, and related, you write that "sampling errors can often be statistically evaluated and corrected" (p8, line 198). While sampling errors are routinely evaluated, I don't think you can say that they can be "corrected". What you can do is express uncertainty in terms of sampling errors, e.g. through confidence intervals, but the sampling error remains and is not "corrected". Perhaps you can say that there are methods to "correctly" take them into account when gauging the degree of uncertainty around estimates.

Response: Thank you for this comment. We agree that further clarification was needed. We have now explained what we mean by data quality (in the introduction), defining good data quality as: low statistical uncertainty and low systematic measurement error and missing data. We argue that both are necessary to derive high-quality insights from DHS data. We also specify early in the introduction that we consider systematic measurement errors/missing data to be the more serious data problem.

In result 3, we make clear that high uncertainties can threaten the predictive power of fitted models. A comparison with measurement errors makes sense because these problems are independent, and data users want to be informed when both occur, because both data characteristics threaten the insights we can derive from analyzing the data. In this sense, our predictions of measurement errors complement the analysis of data quality that is usually provided in the "limitations" sections of scientific studies, which so far have largely been concerned with classical measures of statistical uncertainty (such as standard errors) and have had to limit themselves to acknowledging the potential threat of missing data and measurement errors without precise knowledge in which localities and to which extent they occur.

We finally improved our wording of " ... *taking into account statistical uncertainty...* " in the introduction section.

Please note that your comment was similar to a comment from Reviewer 4, where we responded:

"Thank you for this comment. We agree that the motivation for developing Result 3 could use some clarification. Therefore, we fixed the sentence "...curious to know..." and provided a better argument about why the present study is important for DHS users and for the DHS program. We have added some text to explain this: "Finally, in result 3, we contrasted the spatial distribution of measurement errors generated in result 1 with the statistical uncertainties associated with related public health indicators in the DHS data, which may collectively threaten the quality of insights that can be drawn from DHS data." We added this to emphasize that the motivation for Result 3 is not simply to increase the sample size in the areas where there is higher statistical uncertainty. Instead, we are aiming to define data quality as having both low statistical uncertainty and low bias (i.e., low measurement error and missing data). Result 3 suggests that future data quality assessments can benefit from looking at both dimensions of data quality, not just uncertainty (which can be improved by increasing sample size)."

- That your measures of data quality deteriorate with "remoteness" is perhaps interesting, but remoteness is surely correlated with many other factors, especially literacy / numeracy of respondents. You are careful not to make causal claims about the correlations you document, but on

the other hand it was not clear to me why "remoteness" was chosen as a key correlate of ME.

Response: Thank you for this comment. We added a clearer explanation of why we work with “remoteness” at the beginning of Result 2. The new text reads: *“Remoteness captures a mix of potential factors such as respondent literacy and numeracy, or more challenging conditions for survey collection teams. We did not intend to study these underlying causes. Our aim was to explore the magnitude of this problem and whether it affected the three data quality indicators to a similar extent. In terms of further empirical analysis, the data challenges systematically increasing with remoteness could threaten the insights drawn from the data in two ways. In the best case, measurement errors and missingness are random, leading to more noisy estimates; in the worst case, errors and missingness are systematic, leading to systematic bias in parameter estimates. Data users will need to assess on a case-by-case basis which of these two threats is more relevant to their study based on geographic location, point in time, and variables of interest.”*

- P3, line 49, should be "Practitioners and researchers who work extensively with DHS data..." (drop "rely on"). Line below "data errors can be DISTRIBUTED ACROSS regions" sounds awkward / unclear, please rephrase.

- P4, line 62, should be "for example AT the level of districts"

Response: Thank you for these comments – these issues have been fixed as well.